



# Bacterial and fungal predator - prey interactions modulate soil aggregation

Amandine Erktan[1], Matthias C. Rillig[2], Andrea Carminati[3], Alexandre Jousset[4], Stefan Scheu[1]

[1]J.F. Blumenbach Institute of Zoology and Anthropology, University of Goettingen, 37077 Goettingen, Germany
[2]Institut für Biologie, Freie Universität Berlin, 14195 Berlin, Germany
[3]Chair of Soil Physics, University of Bayreuth, 95440 Bayreuth, Germany
[4]Institute of Environmental Biology, Ecology and Biodiversity, Utrecht University, 3584CH Utrecht, The Netherlands

*Correspondence to*: Amandine Erktan (aerktan@gwdg.de)

## Abstract

The formation and stabilisation of soil macro-aggregates protects soils from erosion, a major worldwide threat on soils. While the role of bacteria and fungi in soil aggregation is well established, how predators feeding on microbes modify soil aggregation has hardly been tested. Here, we studied how predators modulate the effect of microbial prey on soil aggregation. We focused on two predator - prey interactions: bacterial-based interactions comprising amoebae (*Acanthamoeba castellanii*) grazing on free-living bacteria (*Pseudomonas fluorescens*), and fungal-based interactions comprising collembolans (*Heteromurus nitidus*) grazing on saprotrophic fungi (*Chaetomium globosum*). We conducted a microcosm experiment lasting six weeks and assessed changes in soil aggregate formation and stabilisation, together with modifications in soil microbial communities (PLFAs). We further traced the food resource consumed by microbes using $\delta^{13}$C isotopic tracing. The protist *A. castellanii* increased the formation of soil aggregates but decreased their stability, without affecting bacterial abundance and community composition, suggesting that the changes were due to amoebae-mediated changes in the production of bacterial mucilage. Saprotrophic fungi showed the highest positive effect on soil aggregate formation and stabilisation, associated with a more efficient use of particulate organic carbon (chopped litter) added to the microcosms. Adding collembolans decreased the abundance of fungi and their ability to capture carbon of litter origin, with negative consequences on soil aggregation. Our work here has demonstrated that trophic interactions are important for achieving a mechanistic understanding of biological contributions to soil aggregation.



## 1. Introduction

Soil erosion is one of the ten major global threats on soils, leading to dramatic loss of soil biodiversity and a severe decline in agricultural soil fertility (FAO, 2015). Whether a soil is sensitive to water erosion is related to its cohesion between soil particles after contact to water. Soil aggregates are chunks of soils with sizes ranging from micro- to millimetres (Tisdall and

Oades, 1982; Totsche et al., 2018), and their ability to resist to breakdown under drying and wetting cycles is an indicator of soil erodibility (soil aggregate stability; Barthes and Roose, 2002; Norm ISO/FDIS 10930 (E), 2012). High proportions of macroaggregates (> 250 μm) after soil sieving (thereafter referred as aggregate formation), together with a high stability of these aggregates indicate high resistance to hydric erosion.

The stability of soil aggregates has extensively been investigated from a soil management perspective to assess how agricultural

practices influence soil erodibility (Beare et al., 1994; Paul et al., 2013). Most previous studies highlighted that soil organic matter is a key driver of soil aggregate stability (Le Bissonnais et al. 2007; Six et al., 2004; Martens, 2000). As soil organic matter originates from the incorporation of organic debris and root exudates into the soil matrix and its transformation by soil organisms, the essential role of the latter in soil aggregate stability has long been acknowledged. An overall positive role of soil organisms on soil aggregation (aggregate formation and stabilisation) has been highlighted by a recent meta-analysis

(Lehmann et al. 2017). However, soil organisms are very unevenly studied, with few groups, namely bacteria (Watt et al., 1993; Caeser-TonThat et al., 2007), fungi, especially arbuscular mycorrhizae fungi (AMF; Rillig and Mummey, 2006) and earthworms (Bottinelli et al., 2015) receiving most attention.

The mechanisms underlying the effects of bacteria and fungi on soil aggregation have been well described. Briefly, bacteria foster soil aggregation through the production of mucilage, mainly composed of exo-polysaccharides (Bezzate et al. 2000;

Chenu, 1993; Sandhya and Ali, 2014). Bacterial exo-polysaccharides glue soil particles together, resulting in enhanced proportions of soil micro- (<250 μm; Caesar-TonThat et al. 2007) and macroaggregates (>250 μm; Vardharajula and Skz, 2014). Fungi (AMF and saprotrophic fungi) positively influence soil aggregation through the enmeshment of soil particles by fungal hyphae (Degens, 1995; Tisdall and Smith, 1997; Leifheit et al., 2015) and through the enhancement of the cohesion between soil particles by releasing mucilage (Chenu, 1989). Saprotrophic fungi are less studied compared to AMF (Lehmann

et al. 2015) and showed a more transient effect on soil aggregation (Caesar TonThat and Cochran, 2000; Daynes et al., 2012). By contrast, many other soil organisms of microbiota and mesofauna have been largely ignored in respect to their role in soil aggregation (Lehmann et al. 2017), including protists, nematodes, enchytraeids, mites and collembolans. Most of these soil organisms are ubiquitous and abundant. This applies to protists of the genus *Acanthamoeba* (Rodriguez-Zaragoza, 1994) and collembolans (Hopkin, 1997). Both of these groups are known for their important role in processing of soil organic matter

(Stout, 1980; Ekelund and Rønn, 1994; Krashevska et al., 2018; Filser, 2002), the main driver of soil aggregate stability. We thus expect that they play an important role in soil aggregation, but it has never been tested for protists and only few studies exist on collembolans (Siddiky et al., 2012).



Broadening the spectrum of soil organisms studied for their role in soil aggregation is thus needed. However, not only is the diversity of soil organisms important, but their trophic (and non-trophic) interactions need to be considered at close to natural soil conditions. In just one gram of soil up to $10^8$ - $10^9$ bacterial cells, 200 m of fungal hyphae and $10^3$-$10^5$ protists may co-exist and 10,000 – 100,000 individuals of collembolans may be found per square meter (Hopkin, 2007). These figures illustrate that the soil is a densely populated habitat, suggesting that interactions between organisms from various taxa are commonplace. Effects of biotic interactions on soil aggregation have received very limited attention and have been restricted mainly to interactions between fungi and bacteria (Lehmann et al. 2017). Soil microbes are, however, at the basis of the soil food web, comprising a wide range of detritivores and predators, with the latter having major impact on the processing of organic matter (de Vries et al., 2013) and the regulation of the abundance and composition of other biota in soil (Ott et al., 2014, Lang et al. 2014). Considering interactions between various trophic levels thus is needed for a closer understanding on how soil biota affect soil aggregation. Investigating predators feeding on microbes is an ideal starting point. Their influence on soil aggregation may occur (i) through trophic interactions, by modifying the abundance, composition and the activity of the microbes they are feeding on, and (ii) through non-trophic interactions, via the reorganisation of microbiota in the soil matrix, by moving organic matter and microbes. Such reorganisation may occur via feeding activities and through the transport of microbes on the body of their consumers. Evidence of consumers' effects on the growth, abundance and composition of the microbiota through preferential feeding is known for protists (*Acanthoeba*; Fenchel, 1997; Weekers et al. 1995; Rønn et al., 2002) and collembolans (Lussenhop, 1992; Fitter and Garbaye, 1994; Klironomos and Kendrick, 1996). In addition, both protists (*Colpoda*) and collembolans disperse particles attached to their body, namely clay (protists) and microplastics (100-400 µm; collembolans), as well as microbes in experimental systems (Rubinstein et al. 2015; Maaß et al., 2017; Gormsen et al., 2004). How individual predator – prey interactions affect the role of protists and collembolans for soil aggregation, however, has never been studied.

Here, we aimed at investigating how predators modulate the effect of microbial prey on soil aggregation. We focused on two simplified predator-prey systems: a bacterial- and a fungal-based system, representing the two main channels of C fluxes in soil. The bacterial-based system comprised amoebae (*Acanthamoeba castellanii*) grazing on free-living bacteria (*Pseudomonas fluorescens*), and the fungal-based system comprised collembolans (*Heteromurus nitidus*) grazing on saprotrophic fungi (*Chaetomium globosum*). To mechanistically test the effects of these trophic interactions on soil aggregation, we conducted a microcosm experiment over 6 weeks duration and assessed resulting changes in soil aggregate formation and stabilisation, together with modifications in soil microbial communities.



## 2. Material and methods

### 2.1 Experimental design

We re-constructed bacterial-based and fungal-based prey - predator systems in soil microcosms (120 g dry weight of soil per
microcosm). We incubated microbial prey populations, respectively the bacteria *Pseudomonas fluorescens* ($2 \times 10^9$ cfu) and
the saprotrophic fungi *Chaetomium globosum* (4 cm$^3$ of colonized LB agar suspended in autoclaved tap water), both known
for their soil aggregating properties (Caesar-TonThat et al. 2014; Tisdall et al., 2012). We further added their associated
predators: (i) the amoeba species *Acanthamoeba castellanii* ($6 \times 10^8$ individuals), able to perforate bacterial biofilms via the
production of protease (Serrano-Luna et al., 2006; Weekers et al. 1995) and thus potentially inducing modification in bacterial
mucilage, a key agent for soil aggregation; and (ii) the collembolan species *Heteromurus nitidus* (30 individuals), known to
intensively feed on soil fungi (Scheu and Folger 2004). As predators came with their associated microbiota, we accounted for
this "contamination" by inoculating the bacteria *Escherichia coli* ($6 \times 10^8$ individuals), used to culture the amoebae, to the
bacterial (*P. fluorescens*) and control treatments (bacterial-based system; Table 1). Similarly, we added a microbial wash of
equivalent number of collembolan individuals to each microcosm of the fungi (*C. globosum*) and control treatments (fungal-
based system; Table 1). As microbiota inoculation were added in medium (autoclaved tap water with or without smashed LB
agar), the same amounts of medium were added to the respective control treatments. Inoculation was performed in two steps,
with prey added two weeks before their respective predators to allow microbial growth without predator pressure. In addition,
we had a zero control in which only autoclaved tap water was added.

The soil used in the microcosms was a mixture of sand (59.7%), agricultural soil (39.8%) and dried chopped litter (µm to mm
pieces; 0.4%), composed of 85.3% of maize litter and 14.7 % of wheat litter. The agricultural soil was crushed through a 1 mm
sieve to destroy larger aggregates, mixed with sand and litter, and the resulting soil mixture was sterilized by autoclaving (2 h
at 121°C). The properties of the soil mixture (prior litter addition) were: 6.0% clay; 30.8% silt, 13.7% fine sand (63-200 µm),
41.4% medium sand (200-630 µm), 8.1% coarse sand (630-2000 µm), 4.5% CaCO$_3$ and 0.36% organic carbon (analyses were
conducted by LUFA, Speyer, Germany, following the methods VDLUFA I, C2.2.1:2012 and DIN ISO 10694:1996-08). The
agricultural soil was collected from a wheat agricultural experimental field of the University of Göttingen, managed under
conventional tillage and located in the metropolitan area of Göttingen.

### 2.2 Microcosm incubation

Soil microcosms were incubated in a growth chamber at 20°C for a total of 6 weeks (July – August 2017), with the first 2
weeks with microbial prey only. The glass jars (7.5 cm diameter and 10 cm high) containing the soil microcosms (120 g dry
weight of soil mixture with litter) were covered by non-sealed lids, allowing gas and water exchange, but limiting potential
contamination in our non-sterile experimental design. Soil water holding capacity was adjusted weekly to 60% of the maximum
by adding autoclaved tap water in the sterile hood. To monitor the activity of soil organisms, the production of CO$_2$ was



measured twice during the incubation period by titration of 2 mL KOH placed in the jar for 48 h, (Fig. A1). To estimate whether the inoculation treatments modified the soil organic matter content over the incubation period, soil organic carbon (SOC) concentration of aliquots of soil from the treatments was measured after the end of the incubation period (Fig. A2). Aliquots (c.a 500 mg) of dried (105°C, 24 h) and milled (45 s, frequency: 25/s; MM200, Retsch GmbH, Haan, Germany) soils samples were placed at 600°C during 2 h to remove the organic carbon by combustion. Aliquots of 20 mg of milled soil (burnt and unburnt at 600°C) were used to measure their carbon content (Vario EL, Elementar, Hanau, Germany), which represented the inorganic and organic carbon, respectively. The soil organic carbon content was obtained by the difference in the concentrations between the total organic carbon and the inorganic carbon. At the end of the incubation, the presence of amoebae was checked under the light microscope and collembolans were extracted using an entomological exhauster and counted. The survival rate of *H. nitidus* was $83 \pm 21$ %, with $27 \pm 14$ % of juveniles.

**2.3 Soil aggregate properties**

Soil aggregate formation was assessed by soil dry sieving (6 helicoidal movements; 30 cm amplitude) of air-dried (ca. 22°C; 7 days) soil samples, using the following sieves: 10 mm, 5 mm, 3 mm, 2 mm, 250μm and 50μm, resulting in seven diameter classes of aggregates. As soil was crushed through a 1 mm sieve during soil preparation for microcosm incubation, aggregates larger than 1 mm must have been formed during incubation. The term "aggregate formation" is used in general to describe the amount and size of aggregates obtained by dry sieving after incubation. Aggregates are obtained by sieving, which is a way to reveal enhanced cohesion between soil particles. Soil aggregate stability was measured following ISO/FDIS (E) 10930 (2012) described in Le Bissonnais (1996) and Le Bissonnais and Arrouays (1997). Briefly, 8 g of dried (24 h at 40°C) soil aggregates (3 – 5 mm) were gently re-wetted by capillarity for 5 min on a buffer paper lying on a saturated sponge. Aggregates were then transferred into ethanol and aggregates > 50 μm were retrieved by sieving in ethanol. The aggregate fraction > 50 μm was then oven-dried at 40°C for 24 h and sieved using six sieves (2 mm, 1 mm, 500 μm, 200 μm, 100 μm, 50 μm), resulting in seven diameter classes of aggregates. The mean weight diameter (MWD) was calculated as the average diameter of aggregates weighted by the mass proportion of aggregates within each fraction. The MWD of the dry distribution of aggregates, indicating aggregate formation, is noted $MWD_{dd}$. The MWD obtained after dispersion of aggregate by gently rewetting, indicating aggregate stability, is noted $MWD_{as}$.

**2.4 Microbial community composition (phospholipids fatty acids)**

Changes in soil microbial abundance and composition were quantified by phospholipid fatty acid (PLFA) analysis. Lipids were extracted from fresh soil equivalent to 3.5 g dry weight. The soil was frozen at -20ºC after the experiment until further use according to the protocol of Buyer at al. (2012). PLFAs were measured and identified as described in Pollierer et al. (2015) using a gas chromatograph (GC; Clarus 500 with Autosampler, Perkin Elmer, USA). The mass (nmol.g$^{-1}$ of dry soil) of all extracted and identified PLFAs was used as measure of microbial biomass. The PLFA 18:2ω6,9 was used as fungal biomarker





and 8 PLFAs were used as bacterial biomarkers: i15:0, a15:0, i16:0; i17:0 (Gram-positive bacteria), cy17:0, 18:1ω7 (Gram-

negative bacteria), and 16:1ω7 (general bacterial marker) (Frostegård & Bååth 1996; Contosta et al. 2015).

## 2.5 $\delta^{13}$C isotopic signature of soil PLFAs

The isotopic $^{13}$C/$^{12}$C ratios of the PLFAs was measured using a trace gas chromatograph (GC; Thermo Finnigan, Bremen, Germany), equipped with a DB5-DB1 column combination (30 m and 15 m, both 0.25 µm ID, Agilent), and coupled via a GP interface to a Delta Plus mass spectrometer (Thermo Finnigan, Bremen, Germany). The temperature program was run

according to the following steps: 1 min at 80˚C, an increase to 170˚C at a rate of 10˚C/min, an increase to 192°C at a rate of 0.7°C/min, 4 min at 192°C, an increase to 200°C at a rate of 0.7°C/min, an increase to 210°C at a rate of 1.5°C/min, a final increase to 300°C at a rate of 10°C/min, and a final step at 300°C during 10 min. Helium was used as carrier gas for injections (250°C). PLFAs were identified by comparison of their chromatographic retention times with those of standard mixtures composed of 37 different FAMEs  (Fatty Acid Methyl Esters; Sigma Aldrich, St Louis, USA) ranging from C11 to C24 and

26 BAMEs (Bacterial Fatty Acid Methyl Esters; Sigma Aldrich, St Louis, USA). Isotope ratios were expressed vs. Vienna Pee Dee Belemnite standard (V-PDB) as $\delta^{13}$C [‰] = (($^{13}$C/$^{12}$C)$_{sample}$/($^{13}$C/$^{12}$C)$_{standard}$ − 1) * 1000.

The proportion of C from soil (0.36 % of organic C originally present in the wheat agricultural soil) vs. litter origin (0.4 % added chopped litter mainly derived of maize leaf and roots) was calculated using differences in the isotopic signature of the C from the soil and from the added litter. The $^{13}$C/$^{12}$C ratios of the soil and litter were measured using a Delta Plus mass

spectrometer (Thermo Finnigan, Bremen, Germany). Aliquots of ca. 2 g of soil and litter samples were dried (70°C, 24 h), milled (45 s, frequency: 25/s; MM200, Retsch GmbH, Haan, Germany) and placed in a desiccator for 48 h. Aliquots of ca. 25 mg of soil and 0.7 mg of litter were analysed (eight replicates each). The difference between the average isotopic signature of the soil ($\delta^{13}$C$_{Soil}$ = -27.16 ± 0.06 ‰) and the litter ($\delta^{13}$C$_{litter}$ = -13.71 ± 0.06 ‰) covered an amplitude of 13.45 ‰ and set the full range of isotopic variation (100 %) used to define C origin. More precisely, the litter signature was set to 0 % of C from

soil origin and the soil signature to 100 % of C from soil origin. For each treatment and PLFA type (bacterial or fungal markers), the proportion of C of soil origin was calculated as follows: % C$_{soil}$ of x = [($\delta^{13}$C$_x$ - $\delta^{13}$C$_{soil}$) x 100]/ ($\delta^{13}$C$_{litter}$ - $\delta^{13}$C$_{soil}$)]. For bacterial markers, we used average $\delta^{13}$C values weighted by the relative proportion of each bacterial marker considered. As local hotspots of soil can show lower $\delta^{13}$C signal than the average $\delta^{13}$C signature of homogenised soil, the calculation may result in an estimated percentage of C of soil origin higher than 100 %.

## 2.6 Data analyses

Differences between treatments in the degree of soil aggregation, concentration of PLFA markers and proportion of C of soil origin were inspected using generalized least square (GLS) models, followed by ANOVA and post-hoc Tukey tests. Differences in variance between treatments were accounted for in the GLS models. The effects of treatments on bacterial PLFA composition were investigated using non-metric multidimensional scaling (NMDS), followed by discriminant function





analysis (DFA). Overall differences in bacterial PLFA composition within the bacterial and fungal system first were analysed
by MANOVA. Pairwise differences between treatments were further tested using Mahalanobis distances. The relationship
between $MWD_{dd}$ and $MWD_{as}$ and selected explanatory variables, namely the proportion and composition in bacterial PLFA
markers, the proportion of fungal PLFA marker and the concentration in soil organic carbon (SOC) were investigated using
glm models. Only significant models are displayed, unless for the model linking soil aggregation and SOC, provided in

Fig. A2). All the statistical analyses were run separately for the bacterial and fungal systems. Data provided in the text represent
means ± standard deviation. All statistical analyses were conducted in R - version 3.6.1 (R Development Core Team, 2008).

## 3. Results

### 3.1 Soil aggregation as affected by predator - prey system

The 6 week incubation period in microcosms resulted in the formation of aggregates, regardless of the treatment considered

(mean $MWD_{dd}$ across all treatments = 3.84 ± 0.8 mm), compared to the initial soil conditions ($MWD_{dd}$ = 0.77 ± 0.01 mm). The
formed aggregates on average were unstable (mean $MWD_{as}$ across all treatments = 0.58 ± 0.15 mm) and ranged from very
unstable (min. $MWD_{as}$= 0.31 mm) to moderately stable (max. $MWD_{as}$= 1.04 mm), according to the classification of the
international norm ISO/FDIS10930 (E) (2012).

In the bacterial system, inoculation with *P. fluorescens* did not increase the formation of aggregates (Fig. 1A), compared to

the control (CB; *E. coli*). By contrast, *P. fluorescens* significantly increased soil aggregate stability (Fig. 1B) from $MWD_{as}$
(CB) = 0.44 ± 0.03 mm in treatments without *P. fluorescens* to $MWD_{as}$ (B) = 0.57 ± 0.08 mm) in treatments with *P.
fluorescens*. Adding amoebae significantly increased aggregate formation, resulting in the formation of larger diameter
aggregates ($MWD_{dd}$ (B+A) = 4.36 ± 0.64 mm) in treatments with *A. castellanii* compared to the *P. fluorescens* only treatment
($MWD_{dd}$ (B) = 3.45 ± 0.51 mm). By contrast, amoebae suppressed the stabilising effect of bacteria on soil aggregates, indicated

by the lack of significant difference between treatments with bacteria and amoebae ($MWD_{as}$ (B+A) = 0.51 ± 0.08 mm) and the
control (*E. coli* only; $MWD_{as}$ (CB) = 0.44 ± 0.03 mm).

In the fungal system, inoculation with *C. globosum* significantly enhanced the formation and stabilisation of aggregates
($MWD_{dd}$ (F) = 4.8 ± 0.8 mm and $MWD_{as}$ (F) = 0.8 ± 0.15 mm), compared to the control ($MWD_{dd}$ (CF) = 3.5 ± 0.8 mm and
$MWD_{as}$ (CF) = 0.5 ± 0.03 mm). Collembola detrimentally affected these effects by suppressing the positive effect of fungi on

aggregate formation ($MWD_{dd}$ (F+C) = 3.9 ± 0.6 mm) and by reducing the stabilising effect of fungi ($MWD_{as}$ (F+C) = 0.7 ± 0.1
mm). Soil aggregate formation and stability neither differ between the control treatment with water only and the bacterial
treatment (*P. fluorescens*) nor between the former and the control of the bacterial treatment (*E. coli* only), which is consistent
with our non-sterile experimental design. Remarkably, $MWD_{dd}$ (CW) showed a higher variance as in the control of the bacterial
treatment, in which inoculation with *E. coli* reduced the variability.

Regardless of the treatment considered, when higher masses of aggregates (dry sieving) were observed in size classes > 10 mm
(and > 5 mm in the fungal system), lower soil mass were found in the size class < 2 mm (Table A1). Such inversion is logical





as the total mass across the fractions is constant and indicates that very large aggregates > 10 mm (and > 5 mm, respectively) were built upon the soil fraction < 2 mm. Similarly for soil aggregate stability, when higher masses of stable aggregates were found in the classes > 1 mm, we obtained lower amounts of soils in the classes < 0.5 mm. Overall, this indicates that aggregates

> 1 mm have a similar behaviour in terms of resistance to disaggregation in water.

**3.2 Microbial community composition as affected by predator - prey system**

Predator-prey systems had only little influence on soil microbial biomass but modified the composition of microbial communities (Fig. 2). The sum of all bacterial PLFAs, used as an indicator of microbial biomass, did not vary in the bacterial system; and in the fungal system total PLFAs were only increased in the treatments with fungi only (F) and with fungi and

collembolans (F+C) (Fig. 2 A). The sum of total PLFAs was lowest in the control treatment inoculated with water only (CW). The proportion of fungal PLFAs was the highest in microcosms inoculated with fungi (F; % fungal PLFAs = 11.4 ± 2.3 %) and the addition of collembolans significantly reduced their abundance (F+C; % fungal PLFAs = 7.4 ± 1.8 %), which, however, remained significantly higher than in the control (CF; % fungal PLFAs = 3.3 ± 0.6 %). In all other treatments of the fungal and bacterial systems, the proportions of fungal PLFAs were low and/or not variable (Fig. 2 B).

In the fungal system, the proportion of bacterial PLFAs (total and Gram negative) in the treatment fungi alone (F) or fungi and collembolans (F+C) was similar or lower than that in the respective control treatments (CF and CW, Fig. 2 C, D). We did not observe such pattern in the composition of the overall bacterial PLFAs, which significantly distinguish the control with water only (CW) from the three other treatments (F, F+C, CF; Fig. 3 B; Table A2). Even though these three treatments differed little as indicated by Mahalanobis distances, bacterial PLFA composition significantly differed, except between the fungal treatment

(F) and the control of the fungal system (CF) (Table A2). Hence, adding collembolans significantly modified bacterial community composition as compared when only fungi were inoculated (F).  In the bacterial system, the proportion of bacterial PLFAs in microcosms inoculated with bacteria only (B) or with bacteria and amoebae (B+A) were lower than in the respective control treatments (CB, CW; Fig. 2 C). Despite that, the proportion of Gram-negative bacteria was higher these microcosms (B; B+A; Fig. 2 D). This is consistent with the fact that both *P. fluorescens* and *E. coli* are Gram-negative bacteria and these

bacteria had been added to both treatments. The overall composition of bacterial PLFAs also differed between the control treatments (CB, CW) and the treatments with bacteria only (B) or together with amoebae (B+A) (Fig. 3 A).

**3.3 Relationships between soil aggregation, microbial composition and C sources**

In the fungal system, the proportion of fungal PLFAs best explained the formation (MWD$_{dd}$; R$^2$ = 0.29; *P*<0.01) and the stability of soil aggregates (MWD$_{as}$; R$^2$ = 0.23; *P*<0.01) (Fig. 4). Both MWD values (MWD$_{dd}$ and MWD$_{as}$) increased with

increasing proportions of fungal PLFAs across the following treatments: control with water (CW), control of the fungal system (CF), fungi with collembolans (F+C) and fungi only (F) (Fig. 4). In the bacterial system, neither the absolute (total or individual groups) nor the relative composition of bacterial PLFAs significantly correlated with the formation and stabilisation of



aggregates. Generally, the SOC concentration varied little across treatments from 0.6 to 1.3 % and neither was related to
$MWD_{dd}$ nor to $MWD_{as}$ in both the bacterial and fungal systems (Fig. A2).

As indicated by mixing models, most of the bacterial ($73 \pm 14$ % across all treatments) and fungal C ($64 \pm 22$ %) originated
from soil C (rather than litter C). In the bacterial systems these figures did not vary significantly among treatments (Fig. 5).
By contrast, in the fungal systems the origin of fungal and bacterial C differed. In microcosms with fungi only (F), $52 \pm 11$ %
of fungal C originated from soil, whereas $72 \pm 4.3$ % of bacterial C was of soil origin, indicating that fungi captured more litter
C ($48 \pm 11$ %) than bacteria ($28 \pm 4.3$ %). These differences levelled off in presence of collembolans (Fig. 5).

**4. Discussion**

By establishing trophic interactions in microcosms, we assessed short term effects of bacterial grazing by protists and of fungal
grazing by collembolans on soil aggregation. Further, we linked effects of these grazers to changes in microbial community
composition and to the utilisation of two pools of organic matter, i.e. litter and soil organic carbon.

**4.1 Bacteria-based predator - prey system**

Adding the predator *A. castellanii* increased soil aggregate formation and decreased soil aggregate stability. To our knowledge,
this is the first experimental evidence that protists affect soil aggregation. The differential effect on soil aggregate formation
and stability suggests that *A. castellanii* induced the release of compounds which promote soil particle cohesion, but which are
of low water-resistance decreasing aggregate stability. These compounds either may have been released by *A. castellanii* itself
or might have been of bacterial-origin and reflect changes in bacterial composition or activity in presence of *A. castellanii*.

Some protists, such as diatoms, indeed are producing mucilage (Higgins et al. 2002), but evidence for the production of
mucilage by *A. castellanii* is lacking, suggesting that it is more likely that the effect on soil aggregation is mediated by
modifications in the production of bacterial mucilage in response to predation. Bacterial PLFA markers indicated that *A.
castellanii* did not significantly change bacterial abundance and composition. Moreover, the lack of relation between bacterial
PLFA markers and soil aggregation indicates that the effect of *A. castellanii* on soil aggregation was not related to changes in

bacterial community composition. Therefore, the effect of *A. castellanii* on soil aggregation likely was due to the modification
of bacterial activity and thereby of bacterial mucilage production. By grazing on bacteria, protists (notably *Acanthamoeba* sp.)
can drastically reduce biofilm biomass (Weitere et al., 2005) or inversely induce an increased production of bacterial mucilage
(Matz et al., 2004). Enhanced mucilage production is indeed a common strategy used by bacteria in response to predation by
protists, leading to higher bacterial survival and growth (Matz and Kjellberg, 2005; Queck et al. 2006). Such increased

production of bacterial mucilage, notably exo-polysaccharides, may have occurred in our study and could explain the higher
aggregate formation in presence of *A. castellanii*. Remarkably, the soil aggregates formed in presence of protists were less
stable, as compared as when only *P. Fluorescens* was incubated. Reduced aggregate stability may relate to changes in bacterial
mucilage composition or to the release of compounds, such as toxins or metabolites conferring protection to predation and





leading to lower soil particle cohesion as side effect. Modifications in the composition of bacterial mucilage has been observed
under water stress (Vardharajula and Skz, 2014), but likely also is induced by protist predators. In addition, *P. fluoresencens*
respond to grazing by *A. castellanii* through enhanced production of anti-fungal toxins (Jousset and Bonkowski, 2010), and
more generally protists produce bacterial-stimulating metabolites which can modify bacterial activity (Ekelund and Ronn,
1994; Jousset 2008). We propose that changes in bacterial mucilage composition, such as an increase protein/polysaccharides
ratio, or the presence of metabolites trapped in the mucilage network modify its wettability and decrease aggregate stability.
The enhanced stability of soil aggregates in the presence of *A. castellanii* is thus probably not caused by changes in bacterial
exo-polysaccharide production, but rather by the release of non-soluble compounds in the bacterial mucilage, such as proteins
or secondary metabolites reducing wettability as side effect. Further investigations of amount and quality of exo-
polysaccharides produced by *P. fluorescens* in presence of *A. castellanii* is needed for a mechanistic understanding of this
phenomenon.

Both bacterial species used for inoculation (*P. fluorescens* and *E. coli*) increased the formation of macroaggregates. The ability
of *P. fluorescens* to enhance soil particle cohesion is known (Caesar-TonThat et al. 2014) and mainly attributed to the
production of exo-polysaccharides with adhesive properties, notably gellan (Banik et al. 2000). For *E. coli* we did not find
evidence for its potential to increase particle cohesion, but it is likely also related to its ability to produce exo-polysaccharides
(Danese et al. 2000), generally known for their gluing of soil particles (Bezzate et al. 2000; Chenu, 1993; Vardharajula and
Skz, 2014). Furthermore, *P. fluorescens*, but not *E. coli*, increased the stability of soil aggregates. Presumably, this is due to
differences in the composition the extracellular polymeric substances produced by bacteria. Exo-polysaccharides, the main
component of extracellular polymeric substances, alone does not stabilise soil aggregates. Indeed, the positive effect of the
addition of exo-polysaccharides on aggregate stability is maximum well after (2 weeks) the degradation of the polysaccharides
(Martens and Frankenberger, 1992). Moreover, polysaccharides are water-soluble compounds, limiting their role for particle
cohesion in water. Other compounds must be at stake. We know that bacterial mucilage also contains proteins, lipids and
extracellular DNA (Flemming and Wingender, 2010). Some proteins, such as the curli proteins in *E. coli* (Flemming et al.
2016) or hydrophobins in *Bacillus subtilis* (Hobley et al. 2015) enhance the hydrophobicity of bacterial mucilage, as well as
resistance to desiccation (Flemming et al. 2016) and proteins favour aggregate stability (Erktan et al. 2017; Rillig, 2004). In
our bacterial system, we argue that bacterial exopolysaccharide production drove aggregate formation and we suggest that
changes in aggregate stability reflect an enhanced proportion of molecules with hydrophobic domains, such as proteins, in the
bacterial mucilage of *P. fluorescens*.

Notably, aggregate formation and stability in the control (where only water was added) did not differ from the treatments
where *P. fluorescens* and *E. coli* were added. This lack of difference presumably was due to the fact that our systems were not
sterile and therefore the control system with the addition of water only also was colonized by bacteria, as indicated by the lack
of significant differences in total microbial PLFAs in the bacterial systems. In fact, the proportion of bacterial PLFAs was at
a maximum in the control (water) and the treatment where only *E. coli* was inoculated. This indicates that *P. fluorescens*
inhibited the colonisation of the microcosms by bacteria during the experiment. This is supported by the strong differences in





bacterial community composition in treatments with and without *P. fluorescens* as indicated by discriminant function analysis of bacterial PLFAs. Generally, bacteria predominantly used C of soil origin which might have been more accessible to bacteria

than litter C due to their restricted mobility (Tecon and Or, 2017).

## 4.2 Fungi-based predator - prey system

Collembolans reduced the positive effect of *C. globosum* on both aggregate formation and stability. This negative effect was associated with reduced fungal biomass (as indicated by the fungal PLFA marker), which likely was due to the consumption of fungal hyphae by *H. nitidus*. Contrary to this negative effect, Siddiky et al. (2012) showed that collembolans to beneficially

affected soil aggregate stability, which was due to the enhancement of the growth of AMF. These opposite effects presumably are due to the use of different fungi, AMF (Siddiky et al. 2012 and saprotrophic ascomycetes (*C. globosum*; this study). Supporting this conclusion, it is known that collembolans preferentially feed on saprotrophic rather than AMF fungi (Klironomos and Kendrick, 1996; Gange et al. 2000). In fact, Siddiky et al. (2012) assumed that preferential feeding on saprotrophic fungi rather than AMF contributed to enhanced growth of AMF and thereby to AMF-mediated increase in soil

aggregate stability. In our study, only non-AMF was growing in the microcosms and collembolans fed on their hyphae, which led to a negative effect on soil aggregation. Moreover, our study captured the short-term effects of collembolans on soil aggregation, after 4 weeks of incubation, while the study of Siddiky et al. (2012) was run over 14 weeks. Whether the negative short term effect observed here may turn positive in the long term remains to be tested.

Even though not related to soil aggregation, presence of collembolans as well as the "Collembola wash" altered bacterial

community composition suggesting that bacteria associated with collembolans were responsible for these changes. Interestingly, the effect was more pronounced when collembolans were added and not only their wash. One reason for that may be that washing selected only a fraction of bacteria associated with collembolans. Further, dispersion of bacteria through movement of collembolans (Coleman et al., 2002, Gormsen et al. 2004) may have contributed to the more pronounced changes in presence of collembolans. Finally, as collembolans also graze on bacteria (Pollierer et al., 2012), this may have contributed

to changes in bacterial community composition.

The positive effect of *C. globosum* on soil aggregation is consistent with previous studies highlighting the positive and transient effect of non-AMF on soil aggregate stability (Daynes et al., 2012), with a peak at 4 weeks after incubation (Caesar-TonThat, 2000). As expected, the proportion of fungal PLFA markers was highest in the treatment with *C. globosum* only, indicating that our inoculation was successful and triggered fungal development. When *C. globosum* was added, alone or with *H. nitidus*,

the proportion of bacterial PLFAs was reduced, and in parallel the proportion of fungal PLFAs was increased. Also, total microbial biomass was increased slightly presumably reflecting fungal growth. Finally, when incubated alone, *C. globosum* consumed higher proportions of C from the added litter compared to treatments with fungi and Collembolan. The ability to capture litter C likely is linked to the growth of fungal hyphae through air-filled pores (Otten et al. 2001), allowing *C. globosum* to reach a wide range of litter resources. Overall, we demonstrated here for the first time directly, using isotopic labelling, the

ability of saprotrophic fungi to capture litter C resources and that this beneficially affects their ability to increase soil aggregation. Notably, collembolans reduced the use of C from the added litter by fungi. This reduction might have contributed to the decrease in fungal abundance in presence of collembolans. In addition, as collembolans also are feeding on litter (Potapov et al. 2016), competition for litter resources between fungi and collembolans may have resulted in reduced fungal biomass.

## 5. Conclusions

We demonstrated that trophic interactions of predators consuming microbial prey influence soil aggregation. In particular, the protist *A. castellanii* increased the formation of soil aggregates, but decreased their stability. Importantly, these effects were not related to changes in bacterial abundance nor composition, indicating that the effects were mainly caused by changes in microbial mucilage production / composition in response to grazing. In the fungal-based system, the fungal feeding collembola species *H. nitidus* detrimentally affected both aggregate formation and stability. These effects were due to reducing the

abundance of fungi, likely consumed by collembolans. Both bacteria and fungi predominantly incorporated C from soil rather than litter C. However, when fungi (*C. globosum*) were incubated alone, they captured significantly more litter C and this likely favoured fungal growth and thereby soil aggregation. Collembolan counteracted these processes. Overall, the results document that interactions between microorganisms and microbial grazers significantly affect soil aggregate formation with the effect of bacterial grazing strengthening bacteria-mediated aggregate formation while lowering aggregate stability, whereas

grazing on saprotrophic fungi reduced their beneficial effects on soil aggregate formation and stabilisation.

## 6. Data availability

In the case of acceptance of the manuscript, the data supporting the results will be archived in a public repository (Dryad or Zenodo) and the data DOI will be included at the end of the article.

## 7. Author contribution

AE and SS designed the study. MR, AJ and AC provided advise on the experimental design. AE conducted the microcosms experiment. AJ provided the culture of *Acanthamoebae castellanii*. AE analysed the PLFAs data. AE ran the statistical analyses and wrote the first draft of the manuscript, and all authors contributed substantially to revisions.

## 8. Competing interests

The authors declare that they have no conflict of interest.

## 9. Acknowledgements

We are thankful to Theodora Volovei for her help in the set-up of the microcosm experiment and more generally for her technical help and advises. We thank Guido Humpert for conducting the PLFAs extraction and measurement, Zhilei Gao for





preparing the *A. castellanii* culture and Leonie Schardt for conducting the measurement of soil organic carbon concentrations. We thank Dr. Jens Dyckmans for measuring the $\delta^{13}C$ isotopic signal of PLFA samples. This work was supported by the European Commission through the Marie Skłodowska-Curie actions (individual fellowship awarded to Amandine Erktan) within the Horizon 2020 framework [grant number 750249].



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



**Tables**

**Table 1: Experimental design**
Details on the experimental design. X indicate that the prey, predator or associated microbiota has been added to the treatment.

| Treatment | | Bacteria (B) | Bacteria + Amoebe (B+A) | Control bacterial system (CB) | Fungi (F) | Fungi + Collembola (F+C) | Control fungal system (CF) | Control Water |
|---|---|---|---|---|---|---|---|---|
| **Prey** | *Pseudomonas fluorescens* | X | X | | | | | |
| | *Chaetomium globosum* | | | | X | X | | |
| **Predator** | *Acanthamoeba castellanii* | | X | | | | | |
| | *Heteromorus nitidus* | | | | | X | | |
| **Predator associated microbiota** | *Escherichia coli* | X | | X | | | | |
| | Microbial wash of Collembolan | | | | X | | X | |






**Figures**

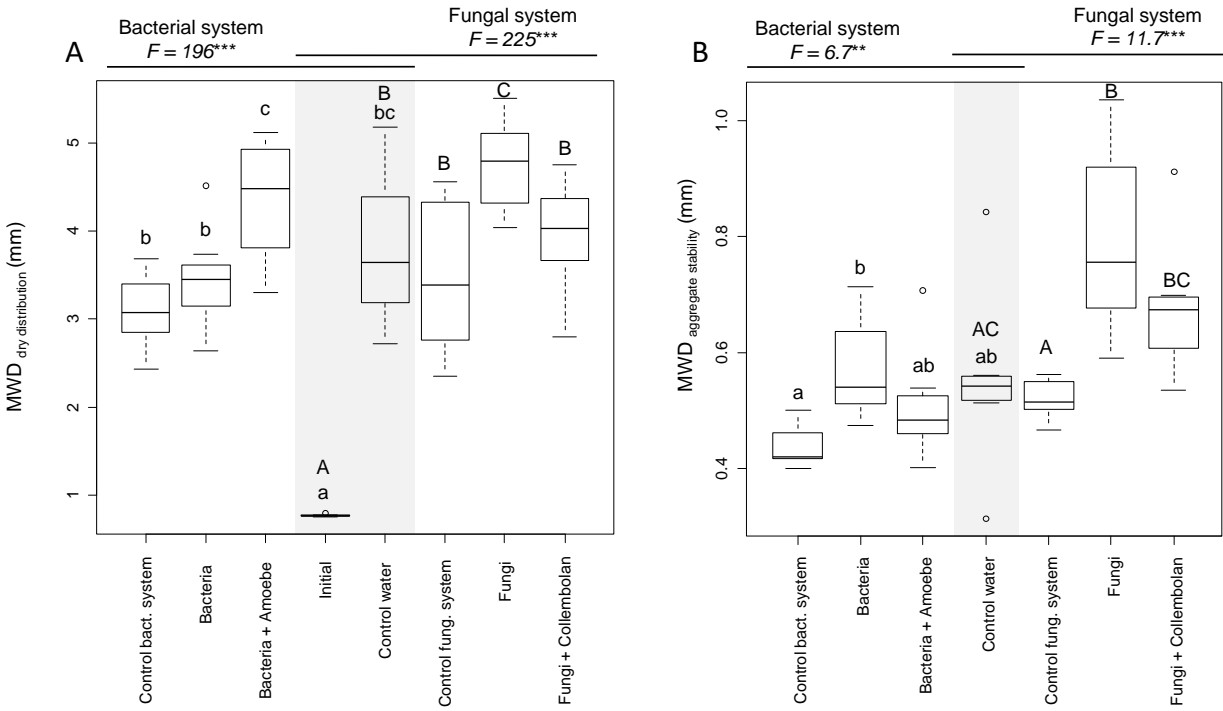

**Figure 1: Effect of bacterial and fungal predator-prey inoculations on (A) soil aggregate formation and (B) soil aggregate stability.** Differences between treatments were tested separately within bacterial and fungal systems using GLS models followed by ANOVA (F; *P*) and post-hoc Tukey tests. Letters (lowercase for bacterial system and capital for fungal system) indicate significant differences between treatments according to Tukey tests. Grey background indicates control treatments common to bacterial and fungal systems. Significance levels are: ***$p < 0.001$; **$p < 0.01$.






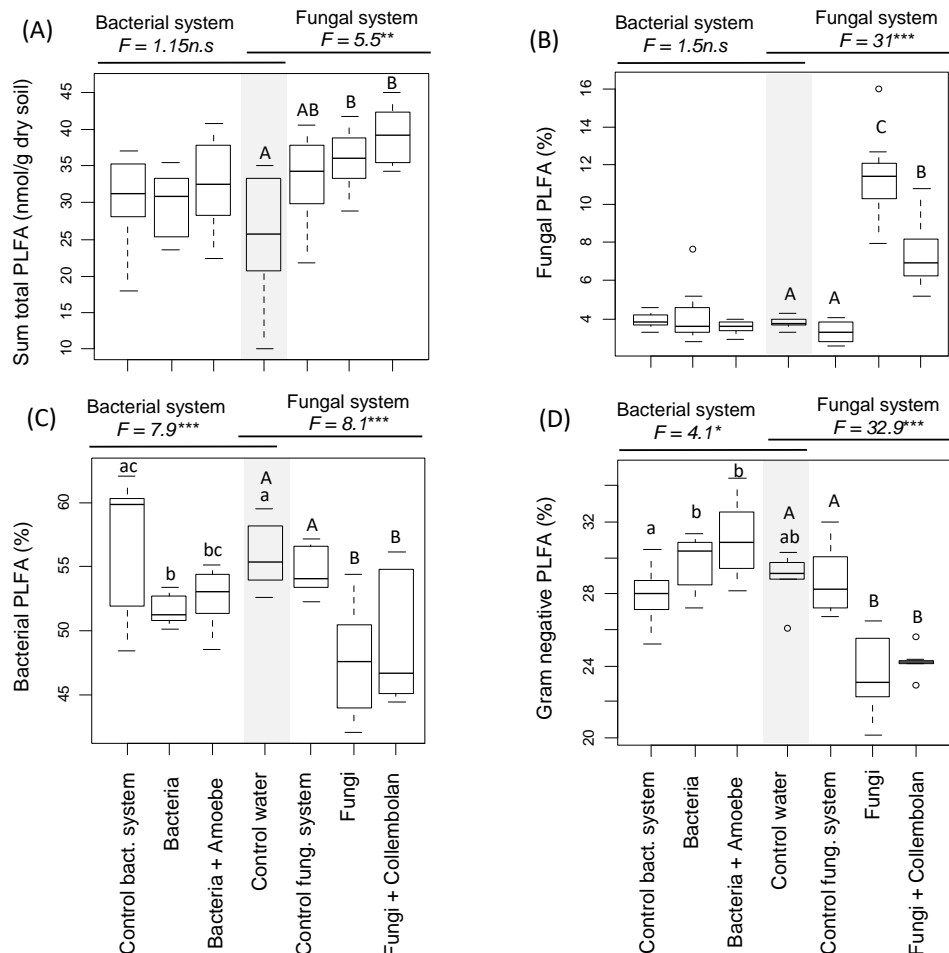

**Figure 2: Quantities and proportions in bacterial and fungal PLFA soil markers.** Details are: (A) sum of total PLFA and proportions of (B) bacterial markers. (C) Gram negative markers and (D) fungal marker. Differences between treatments were tested separately within bacterial and fungal systems, using GLS models, followed by ANOVA (F; *P*) and post-hoc Tukey tests. Letters (lowercase for bacterial system and capital for fungal system) indicate significant differences between treatments according to Tukey tests. Grey background indicates control treatments common to bacterial and fungal systems. Significance levels are: \*\*\*$p < 0.001$; \*\*$p < 0.01$; \*p $< 0.05$; n.s $> 0.05$.





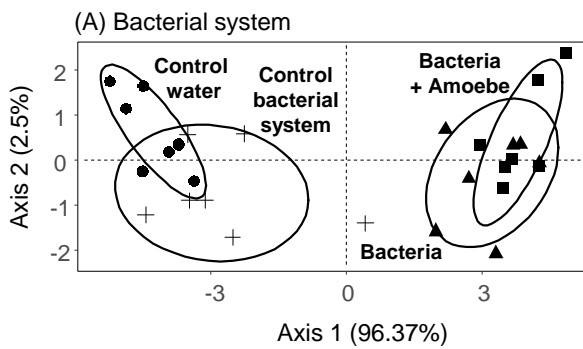

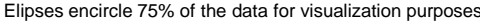

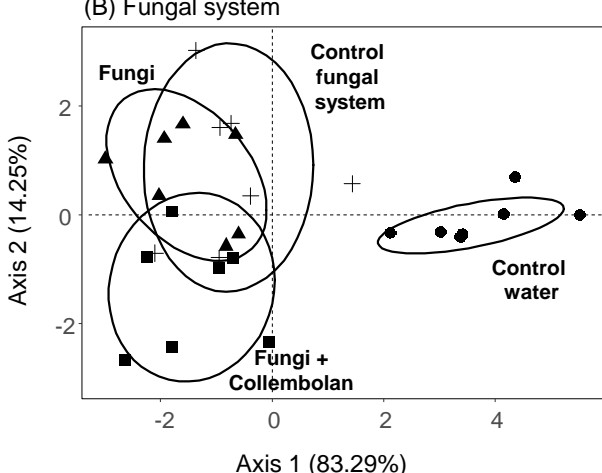

**Figure 3: Composition in bacterial PLFA markers.** Discriminant function analysis of the bacterial PLFA markers in the (A) bacterial and (B) fungal systems. Differences between treatments were tested using NMDS, followed by MANOVA and Mahalanobsis distances. Details of the results of these statistical tests are provided in Table A2. Ellipses encircles 75 % of the data for visualisation purposes.






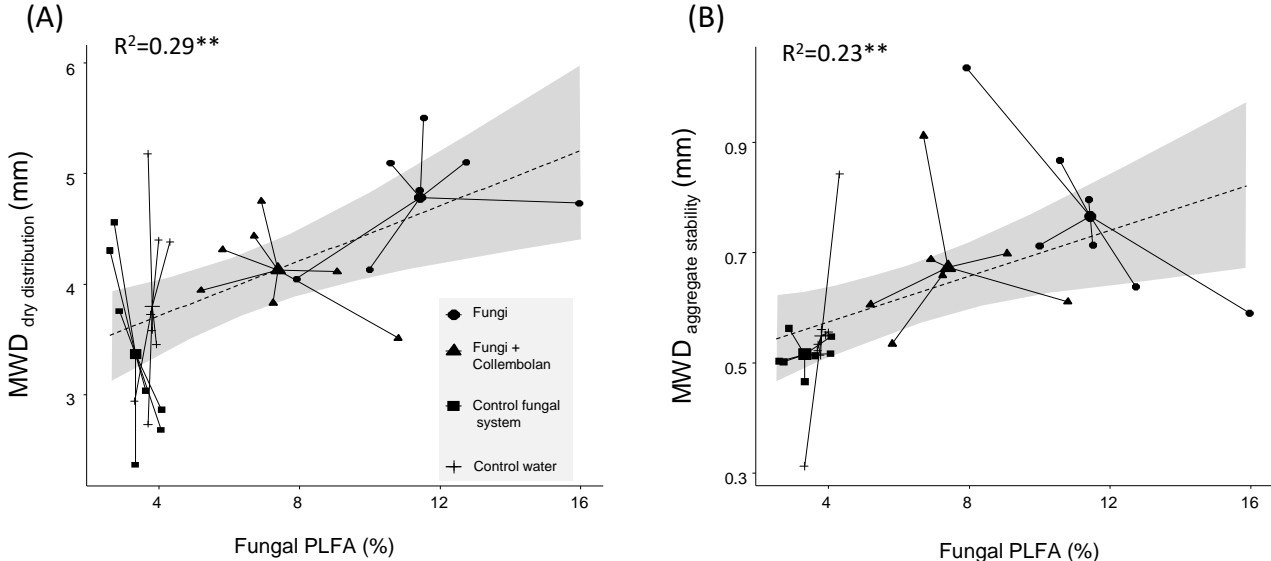

**Figure 4: Relationships between fungal PLFA markers and soil aggregation:** (A) soil aggregate formation and (B) soil aggregate stability. Relationships were investigated using generalised linear models (glm; $R^2$, $P$). The dashed lines represent
the relations according to the models and the grey regions represent the confident intervals (95 %) around the regressions. Significance level is **$p < 0.01$.







**Figure 5: Relative contribution of C from soil over litter origin in bacterial and fungal PLFA markers.** Differences
between treatments were tested separately within bacterial and fungal systems, using GLS models, followed by ANOVA (F;
*P*) and post-hoc Tukey tests. Letters indicate significant differences between treatments according to Tukey tests. For the four
treatments of the fungal system, the differences in C origin in bacterial and fungal PLFA were similarly tested (pairwise
comparison. F; *P*). Grey boxes indicates fungal PLFAs. Significance levels are: \*\*\**p* < 0.001; \*\**p* < 0.01; \*p < 0.05; n.s >
0.05.



**Appendice**

| | $F$ | $P$ | Bacterial system | | | | | Fungal system | | | $P$ | $F$ |
|---|---|---|---|---|---|---|---|---|---|---|---|---|
| | | | Control bact. System | Bacteria | Bacteria + Amoebe | Initial | Control water | Control fung. System | Fungi | Fungi + Collembolan | | |
| **Soil aggregate formation (% in dry mass of soil aggregate per size class)** | | | | | | | | | | | | |
| Permanova | 57 | *** | R2 = 0.87 | | | | | R2 = 0.84 | | | *** | 47 |
| >10 mm | 7.9 | *** | 8.7±1.7 a | 10.2±3.6a | 17.1±4.5b | - | 13±5.7 aBC | 11.4±4.8B | 17.7±4 C | 14.2±3.7 BC | † (0.07) | 2.6 |
| 5 - 10 mm | 2.1 | n.s | 12.7±3.1 | 14.2±2.7 | 16.5±2.8 | - | 14.9±2.3 aB | 13.3±6.5B | 21.6±3.C | 15±3.9 B | *** | 8.4 |
| 3 - 5 mm | 1.8 | n.s | 8.7±2.1 | 10.4±1.1 | 10.7±1.2 | - | 10.1±1.7 | 9.6±2.4 | 11±1.6 | 10.7±2.2 | n.s | 0.7 |
| 2 - 3 mm | 1.1 | n.s | 5.9±1 | 7.1±1.4 | 6±0.8 | - | 6.2±0.9 | 6.4±1.5 | 5.4±0.7 | 6.9±2.2 | n.s | 2.1 |
| 0.25 – 2 mm | 47 | *** | 52.5±5.4b | 48.9±5 bc | 42.6±6 c | 64±1.1 aA | 48±7.3 bcB | 50.2±8.1B | 38.1±3.C | 46.3±6.6 B | *** | 89 |
| 0.05-0.25 mm | 841 | *** | 9.9±1 b | 7.9±1.5 c | 6.1±1.1 c | 30.1±0.8 aA | 7.1±2 cBC | 8±1.4 B | 5.6±0.9 C | 6±1.8 BC | *** | 897 |
| < 0.05 mm | 137 | *** | 1.6±0.4 b | 1.2±0.4 bc | 0.9±0.2 c | 5.9±0.5 aA | 0.7±0.4 bBC | 1.2±0.3 B | 0.7±0.3 C | 0.9±0.2 BC | *** | 140 |
| **Soil aggregate stability (% in dry mass of water-resistant soil aggregate per size class)** | | | | | | | | | | | | |
| Permanova | 1.9 | † (0.06) | R2 = 0.18 | | | | | R2 = 0.29 | | | ** | 4.0 |
| > 2mm | 5.0 | ** | 1.1±0.7 a | 3.7±2 b | 2.4±1.7 ab | - | 3.5±3.2 abAC | 2.4±0.8 A | 9.7±3.6 B | 6.3±3.1 BC | *** | 12 |
| 1 – 2 mm | 14 | *** | 2±0.9 a | 5±1.6 b | 3.4±2.1 ab | - | 4.9±0.9 bA | 4.2±0.8 A | 7.3±2.3 B | 6.4±1 B | *** | 8.8 |
| 0.5 – 1 mm | 2.2 | n.s | 24.7±1.6 | 26.7±1.7 | 26.9±2.2 | - | 24.8±8.8 | 26.5±1.5 | 25.9±1.6 | 27.1±4.4 | n.s | 0.3 |




| | | | | | | | | | | | | |
|---|---|---|---|---|---|---|---|---|---|---|---|
| 0.2-0.5 mm | 12 | *** | 44.4±1.1a | 40.7±2.1b | 41.7±2.3b | - | 40.4±1.7 bC | 42.8±1.4A | 36.1±4 B | 38.4±3.6 BC | *** | 8.4 |
| 0.1-0.2 mm | 4.8 | ** | 13±0.8 a | 12.1±1.1ab | 11.7±1.4ab | - | 11.3±0.9 | 11.8±0.5 | 10.4±1.5 | 10.9±1.2 | n.s | 2.5 |
| 0.05-0.1 mm | 2.6 | † (0.07) | 6.6±0.8 a | 5.2±1.2 b | 6.3±0.7 ab | - | 5.6±1.5 | 5.4±0.8 | 4.7±1.3 | 5.1±0.8 | n.s | 0.6 |
| < 0.05 mm | 2.0 | n.s | 8.2±1.6 | 6.7±1 | 7.6±1.6 | - | 9.3±8.1 | 6.9±1.6 | 5.7±0.9 | 5.8±0.5 | n.s | 1.6 |

**Table A1: Effects of bacterial and fungal predator-prey interactions on aggregate fractions.** The upper part displays aggregate fractions obtained by dry sieving. The lower part displays water stable aggregate fractions (dry mass) obtained after capillarity rewetting, followed by drying. For each predator-prey system and aggregate distribution, differences between distributions of fractions were tested by PerMANOVA. For each aggregate fraction, differences between treatments were tested separately for the bacterial and fungal system using GLS models followed by ANOVA and post-hoc Tukey tests. Letters (lowercase for bacterial system and capital for fungal system) indicate significant differences between treatments according to Tukey tests; ***$p < 0.001$; **$p < 0.01$; †$p < 0.08$; n.s $> 0.08$




| Pair of treatments (a // b) | Statistical parameters | | |
| --- | --- | --- | --- |
| | F | *P* | Mahalanobsis distance |
| **Bacterial system** | | | |
| Overall differences (Manova) | 16 | *** | - |
| Bacteria // Bacteria +Amoebae | 1.6 | n.s | 1.5 |
| Bacteria // Control bact. system | 21 | *** | 3.2 |
| Bacteria // Control Water | 113 | *** | 4.9 |
| Bacteria + Amoebae // Control bact. system | 32 | *** | 4.3 |
| Bacteria + Amoebae // Control Water | 178 | *** | 5.1 |
| Control bact. System // Control Water | 3.0 | n.s | 2.0 |
| **Fungal system** | | | |
| Overall differences (Manova) | 12 | *** | - |
| Fungi // Fungi + Collembolan | 5.4 | * | 2.9 |
| Fungi // Control fung. system | 2.6 | n.s | 1.2 |
| Fungi // Control Water | 29 | *** | 5.3 |
| Fungi + Collembolan // Control fung. system | 4.0 | * | 3.3 |
| Fungi + Collembolan // Control Water | 33 | *** | 5.6 |
| Control fung. system // Control Water | 18 | *** | 4.3 |

***P<0.001; *P<0.05; n.s: P>0.05

15 **Table A2: Composition of bacterial PLFA markers**
Differences between treatments in terms of bacterial PLFAs composition were analysed using non-metric multidimensional analysis (NMDS), followed by discriminant function analysis (DFA). In particular, overall differences in terms of bacterial PLFAs composition within the bacterial and fungal systems were first tested by Manova (F; *P*). Pairwise differences between treatments were further investigated using Mahalanobsis distances (F; *P*). Significance levels are: ***p < 0.001; *p < 0.05;
20 n.s: p>0.05.



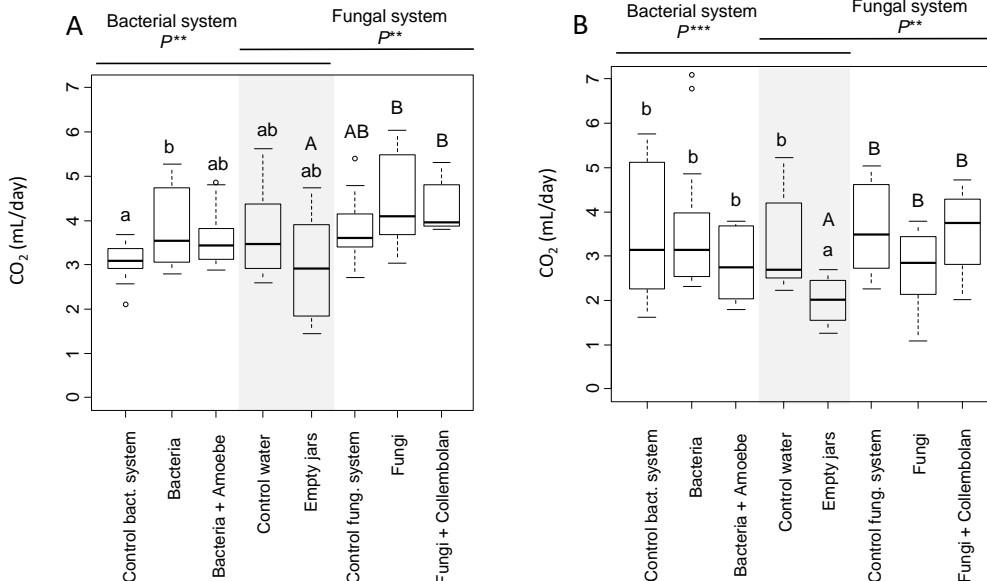

**Figure A1: Production of CO₂ by soil microbes and animals**, after four (A) and six weeks of incubation (B). Differences between treatments were tested separately within bacterial and fungal systems using GLS models, followed by ANOVA and post-hoc Tukey tests. Letters (lowercase for bacterial system and capital for fungal system) indicate significant differences between treatments. Grey background indicates control treatments of both the bacterial and fungal system; $***p < 0.001$; $**p < 0.01$.





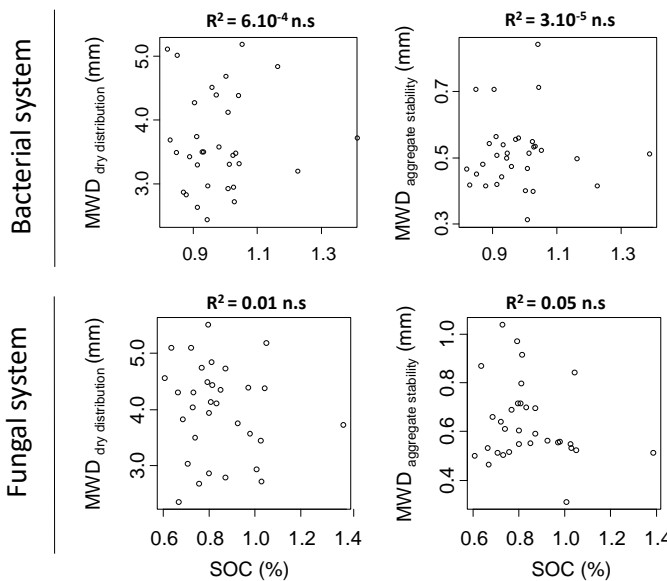

**Figure A2: Relationships between SOC and soil aggregation.** Soil aggregate formation (left panel) and soil aggregate stability (right panel). Relationships were investigated using glm models; ns, P>0.05