# Peer review of "Protists and collembolans alter microbial community composition, C dynamics and soil aggregation in simplified consumer - prey systems"

_Biogeosciences, 2020_

## Referee Comment (RC1) · Anonymous Referee #1 · 16 Mar 2020

The paper is overall interesting and covers a novel aspect in how soil aggregation can appear. Testing two different species that are located in two very distinct taxonomic groups of soil biodiversity and that represent diverse functional groups is nice and can provide a model system to observe some changes. The writing is good but could be a bit more concise in the introduction. Of course, these do not represent even a fraction of the full taxonomic and functional diversity of all soil organisms so it might be that the findings might not represent the function of most other taxa. This is fine but it would be good to acknowledge. Overall the idea behind the experiment are really interesting and relevant. But many aspects as shown below make me wonder if most of the conclusions can actually be drawn. . . Several things should be done to actually make this study publishable that I highlight below. It seems very important to investigate the feed-

ing preference of the two species used. Many Acanthamoeba can indeed also feed on fungi (yeasts and spores) so it seems crucial to check if they feed only on bacteria or if they also change fungi. It should also be thoroughly checked if the prey is a good one to model all representative microbes (either bacteria or fungi)- Pseudomonas is often not the preferred prey of Acanthamoeba species. Please test this interaction in simple microcosm experiments to show that the protists grow on the bacteria. Furthermore, LB agar was used in the fungal treatment that is super beneficial for bacterial growth. How certain is it that fungal-feeding alone cause the observed effects or if it is a mix of bacteria/fungal effects? Some follow up tests are needed to confirm the feeding interactions of the collembola also with bacteria, in interaction with bacteria and fungi etc. the PLFA data provided are nice and partly cover some of the issues but still it is not ensured that competition between inoculated bacteria with other bacteria or between bacteria and fungi are not the actual cause of the experiments. Overall so far it can be said that something happened with predators but I doubt we can really link it to the suggested feeding interactions or energy channels. Please adjust and ideally follow up with some confirmative experiments. Please report more information on specific PLFAs to provide an overview on other bacteria that were potentially contaminating the system. The focus is only on some aspects in the collembola treatment. According to the analyses of gram negative bacteria, it seems that these increase with protists potentially suggesting that the inoculated Pseudomonas are not preyed upon. Please clarify and, as mentioned above, provide an experiment to show successful predation of amoeba on the exact Pseudomonas strain used here (please also report this one). Regarding the discussion, please adjust based on the changes in the results and comments above as I think many claims cannot be made with the current results (e.g.2nd line in discussion and the following- protists do not show a significant effect on aggregation). Similarly, a major discussion in the fungal system is based on a non-significant effect. As long as we use post-hoc statistics, this is determines our findings and what we should be reported. A repetition of the experiment with more replications could be done to increase the statistical power. Overall, throughout the discussion,

conclusions and abstract, many major points are not supported by results and should be adjusted. In fig.1 it is shown that there is no significant difference in the aggregate stability in bacteria alone and bacteria with protist treatments. The same holds for the fungal treatment. As such, this should not be reported as an effect in several parts of the text! I wonder if the composition analysis done in Fig.3 makes sense... In the experiment you would expect a single PLFA marker in the fungal treatment (E.coli) and two in the bacterial treatment (E.coli and Pseudomonas- or even one as both are gram-negative). This figure shows that the setup seems very contaminated which make all results obtained little reliable. Also, I wonder what the relevance for these super simplistic approaches are as many things like interactions that cause biofilm production or integration of bacteria and fungi (and algae) into more stable structures might be needed to make ecological sense of aggregate formation? Please update the references and include some more recent references as only few are from 2018 and none from 2019 or 2020. A lot of work on soil biodiversity, especially on protists including interactions with bacteria, has been done in the last years- even including some papers by the authors that would bring the writing into a more novel context.

Minor comments L238: was higher IN these...

---

## Referee Comment (RC2) · Anonymous Referee #2 · 17 Mar 2020

General: In general I find this study to be well executed and of interest. Including trophic structure in our assessments of soils is an important and understudied topic.

While most of the work is well executed, the authors spend a large portion of their discussion talking about bacterial mucosal production, but this is never actually tested. If a huge portion of the work depends on understanding how trophic structure influences bacterial mucosal production, then it would be important that this is assessed. I would be hesitant to focus so intently on this interpretation, and spend more time addressing the various components you did test.

Additionally, I believe that the 13C portion of this analysis to determine differences in soil and litter derived C needs to be expanded on. This could be an important conclusion, but it is unclear how this work was done, and whether or not labeled litter

was added.

Abstract: No specific comments.

Intro: In general, this is a well written introduction. The authors lay out multiple factors on how microbes and mesofauna influence soil aggregation. It is at times a bit repetitive though, consistently focusing on the lack of trophic structure assessment to soil aggregation.

Additionally, I believe that the authors focus on soil aggregation limits the scope of this study. The authors are assessing multiple components of the soil environment, and therefore, it would be ideal if they could expand their introduction of topics beyond soil aggregation. The authors explore the influence of trophic interactions on soil microbial community formation and on the incorporation of C and on $CO_2$ emissions. If the authors were more concise, they would have room to include additional dimensions to their work.

Methods:

Line 101: Can you clarify this detail a bit more? I think the point is that when you add mesofauna they introduce new microbial organisms, and to account for this you also added microbes to the control treatments, but I am not entirely clear on this detail. How did you detail the Predator associated microbiota?

I find the 13C-12C comparison protocol confusing. Could you expand your discussion of how you are able to differentiate between soil and litter sources? In particular, how are you assessing the final amount of 13C in your soils. Are you obtaining this information from GCMS work, or are you specifically measuring them using an isotopic analysis device? Additionally, I am unclear as to how you are able to ultimate differentiate whether the 13C in you sample came from litter or soil, unless you inoculated with 13C labeled litter.

Results:

Are the control treatments truly just E. coli? I presume that because they were made from field soil, there is also a natural microbial community. This is not necessarily a problem, but if you are labeling these as E. coli only, that may be misleading.

Line 211: Awkward phrasing, maybe adjust to "Neither soil aggregate formation nor stability differed" and break this sentence up into two different sentences.

Lines 210-214: This paragraph starts with fungal results, but then also addresses other treatments. Maybe split this into two paragraphs, as it is difficult o follow the portion of the results in the second half of this paragraph.

Discussion:

While PLFA is an acceptable method, its ability to measure more fine scale changes in community composition is limited. It is possible that changes did occur, but they were not obvious with PLFA analysis.

Is it possible that your soils were water limited prior to the experiment, and that by adding water to the system, that alone was responsible for helping stabilize the soil aggregates?

Line 223: missing a )

Is the collembolan species used known to also feed on bacteria? If so, how would this influence the results?

Why are the $CO_2$ respiration amounts not mentioned throughout the study? It seems like this would be of interest considering that these metrics are often used to estimate microbial biomass.

Additional literature to consider including:

Bradford, M.A., 2016. Re-visioning soil food webs. Soil Biology and Biochemistry 102, 1–3.

[Figure]

Bailey, V.L., Fansler, S.J., Stegen, J.C., McCue, L.A., 2013. Linking microbial community structure to $\beta$-glucosidic function in soil aggregates. The ISME journal 7, 2044.

Crowther, T.W., Thomas, S.M., Maynard, D.S., Baldrian, P., Covey, K., Frey, S.D., van Diepen, L.T.A., Bradford, M.A., 2015. Biotic interactions mediate soil microbial feedbacks to climate change. Proceedings of the National Academy of Sciences 112, 7033-7038.

Grandy, A.S., Wieder, W.R., Wickings, K., Kyker-Snowman, E., 2016. Beyond microbes: Are fauna the next frontier in soil biogeochemical models? Soil Biology and Biochemistry 102, 40-44.

Jiang, Y., Liu, M., Zhang, J., Chen, Y., Chen, X., Chen, L., Li, H., Zhang, X.-X., Sun, B., 2017. Nematode grazing promotes bacterial community dynamics in soil at the aggregate level. The ISME Journal 11, 2705-2717.

Lucas, J.M., McBride, S., Strickland, M.S.S., 2020. Trophic level mediates soil microbial composition and function. Soil Biology and Biochemistry.

---

## Author Comment (AC1) · 28 Apr 2020

We thank the reviewers for the very detailed comments which helped us to improve our manuscript. The response to comments from the two referees is organized according to the Biogeosciences's guidelines as follows: (1) comments from Referees, (2) author's response, (3) author's changes in manuscript. We did not directly quote updated text nor the exact location of the changes made in the updated manuscript as co-authors are still working on it and minor changes can still occur. According to the guidelines of Biogesociences, we submit the answer to the comments from the two referees first and then the updated manuscript.

Anonymous Referee #1 (1) The paper is overall interesting and covers a novel aspect in how soil aggregation can appear. Testing two different species that are located in two very distinct taxonomic groups of soil biodiversity and that represent diverse functional groups is nice and can provide a model system to observe some changes. The writing is good but could be a bit more concise in the introduction. (2) We thank you for the positive comments. We agree that the introduction is sometimes repetitive. (3) We re-wrote most of the introduction. In particular, we broaden the scope of our study to the effects of predator-prey interactions on microbial community composition, C dynamics and soil aggregation. By doing so, the part on soil aggregation was reduced and repetitions were avoided.

(1) Of course, these do not represent even a fraction of the full taxonomic and functional diversity of all soil organisms so it might be that the findings might not represent the function of most other taxa. This is fine but it would be good to acknowledge. (2) We agree that our simplified system does not capture the complex interactions occurring in soils. However, we believe that microcosm experiments with simplified interactions are valuable tools to decipher representative mechanisms which occur in more complex systems, but can't be directly studied in real soil system because of their complexity. (3) We made clearer that the simplification of our experimental system is a limitation and made sure not to abusively generalize results.

(1) Overall the idea behind the experiment are really interesting and relevant. But many aspects as shown below make me wonder if most of the conclusions can actually be drawn... Several things should be done to actually make this study publishable that I highlight below. (2) We thank you for pinpointing the interest of our study as well as for your valuable comments. Please, see our point by point response for an in-depth response to your comments.

(1) It seems very important to investigate the feeding preference of the two species used. Many Acanthamoeba can indeed also feed on fungi (yeasts and spores) so it seems crucial to check if they feed only on bacteria or if they also change fungi. (2) Acanthamoeba has indeed been reported to be a generalist consumer under lab

conditions. In our experimental setup, however, we did not observe a significative change in fungal PLFA markers in response to the addition of A. castellanii (Figure 1 B). The amount of fungal PLFA markers did not vary in the bacterial system, and was constantly low. We thus conclude that A. castellanii did not significantly feed on fungi in our system.

(1) It should also be thoroughly checked if the prey is a good one to model all representative microbes (either bacteria or fungi)- Pseudomonas is often not the preferred prey of Acanthamoeba species. Please test this interaction in simple microcosm experiments to show that the protists grow on the bacteria. (2) In fact, we know that Acanthamoeba castellanii feeds on Pseudomonas fluorescens, but they prefer non-toxic strains (Jousset et al. 2009). In our study, we used exactly the same wild-type strain as in the study of Jousset et al. (2009), who showed that A. castellanii preferably feed on non-toxic strains (signal blind, non-toxic gacS-deficient mutants of P. fluorescens), but also fed to a lesser degree on the wild-type P. fluorescens. In our study, we selected P. fluorescens (wild-type) as it often occurs in high frequency in soils (Dubuis et al., 2017), produces mucilage (we checked visually with the ink method) and is known for its soil aggregating properties (Caesar-TonThat et al. 2014). Moreover, P. fluorescens is known to react to consumption by A. castellanii by producing antibiotic phenolic compounds (Jousset and Bonkowski, 2010), which can modify the soil microbial community (Jousset et al., 2010). Besides, phenolic compounds recently also have been proven to be involved in soil aggregation (Yoshikawa et al. 2018), but no links to protist-bacteria interactions were made. We also expected that P. fluorescens modulates mucilage production in response to protozoan consumption, as it is a common strategy of bacteria in response to protist predation (Matz and Kjellberg, 2005; Queck et al. 2006), with expected consequences on soil aggregation because mucilage is playing a central role in soil particle cohesion. Caesar-TonThat, T. C., Stevens, W. B., Sainju, U. M., Caesar, A. J., West, M., Gaskin, J. F. 2014. Soil-Aggregating Bacterial Community as Affected by Irrigation, Tillage, and Cropping System in the Northern Great Plains. Soil Sci., 179 (1), 11-20. Dubuis, C., Keel, C., Haas, D. 2007. Dialogues of root-colonizing biocontrol pseudomonads. European Journal of Plant Pathology 119, 311-328 Jousset, A., Rochat, L., Péchy-Tarr, M., Keel, C., Scheu, S., Bonkowski, M. 2009. Predators promote defence of rhizosphere bacterial populations by selective feeding on non-toxic cheaters. The ISME journal, 3(6), 666-674. Jousset, A., and Bonkowski, M. 2010. The model predator Acanthamoeba castellanii induces the production of 2, 4, DAPG by the biocontrol strain Pseudomonas fluorescens Q2-87. Soil Biol. Biochem., 42(9), 1647-1649. Matz, C., Kjelleberg, S. 2005. Off the hook–how bacteria survive protozoan grazing. Trends microbiol., 13(7), 302-307. Queck, S. Y., Weitere, M., Moreno, A. M., Rice, S. A., Kjelleberg, S. 2006. The role of quorum sensing mediated developmental traits in the resistance of Serratia marcescens biofilms against protozoan grazing. Env. Microbiol., 8(6), 1017-1025. Yoshikawa, S., Kuroda, Y., Ueno, H., Kajiura, M., Ae, N. 2018. Effect of phenolic acids on the formation and stabilization of soil aggregates. Soil Science and Plant Nutrition, 64(3), 323-334. (3) We specified in the material and methods why we chose P. fluorescens and A. castellanii. We refer to the study of Jousset et al. (2009) and note that P. fluorescens is usually a toxic non-preferred prey, but also is consumed by A. castellanii. In the discussion, we now discuss how the choice of P. fluorescens may have driven the effects of A. castellanii on soil aggregation in our system.

(1) Furthermore, LB agar was used in the fungal treatment that is super beneficial for bacterial growth. How certain is it that fungal-feeding alone cause the observed effects or if it is a mix of bacteria/fungal effects? (2) We agree that the effect of bacteria often complement the one of fungi (Aspiras 1971; Bonfante and Anca 2009). In our study, we did not observe an increase in bacterial PLFA markers when LB agar was added, alone in the "remaining microbial background + Collembolan wash" treatment or with C. globosum in the "remaining microbial background + Collembolan wash + C. globosum" treatment as shown in Figure 1 C. As a consequence, we conclude that the addition of LB agar did not induce a significant increase in bacterial growth in our study, and did not function as major driver of soil aggregation. Aspiras, R. B., Allen, O. N., Harris, R. F., and Chesters, G. (1971). Aggregate stabilization by filamentous microorganisms.

Soil Sci. 112, 282–284. Bonfante, P., Anca, I.-A. (2009) Plants, mycorrhizal fungi, and bacteria: A network of interactions. Annual Review of Microbiology 63, 363-383 (3) In the discussion, we point out more clearly how the interactions between bacteria and fungi could be the main reason for changes in soil aggregation.

(1) Some follow up tests are needed to confirm the feeding interactions of the collembola also with bacteria, in interaction with bacteria and fungi etc. (2) It has been shown in previous experiments that H. nitidus feeds on C. globosum and also reproduces when fed with this fungal species (Pollierer et al. 2019). We now refer to this paper in the manuscript. In the same study it has been shown that H. nitidus feeds as well on bacteria, notably Pseudomonas fluorescens. However, when fed with bacteria, H. nitidus did not reproduce indicting that bacteria are of lower food quality than C. globosum. These results also nicely illustrate the concept of food flexibility (Briones et al., 2018) indicating that soil animals often prefer certain food resources but also feed on other resources if the preferred resources are absent. We now explicitly refer to these studies when discussing our findings. Briones, M.J.I. (2018) The serendipitous value of soil fauna in ecosystem functioning: The unexplained explained. Front. Environ. Sci. 6:149. doi: 10.3389/fenvs.2018.00149 Pollierer, M. M., Larsen, T., Potapov, A., Brückner, A., Heethoff, M., Dyckmans, J., & Scheu, S. (2019). Compound‐specific isotope analysis of amino acids as a new tool to uncover trophic chains in soil food webs. Ecological Monographs, 89(4), e01384 (3) We specified in the material and methods the feeding preference of H. nitidus for C. globosum. In the discussion, we stress that the changes in the bacterial community composition after the addition of H. nitidus may also have been due to consumption of bacteria.

(1) the PLFA data provided are nice and partly cover some of the issues but still it is not ensured that competition between inoculated bacteria with other bacteria or between bacteria and fungi are not the actual cause of the experiments. (2) We agree we overlooked the effects of microbial competition, or more generally interactions between the remaining microorganisms in the microcosms and the added bacterial and

fungal strains. We fully agree that this is an important aspect to understand variations in the composition of the microbial community, and how this can relate to soil aggregation. (3) We consider these interacting microbial processes in the revised version of the manuscript. We renamed the all the treatments to show that a remaining microbial background was present in our systems because of our non-sterile set-up. The treatment with E. coli thus was not only colonized by E. coli, but contained the remaining microbial background and E. coli. We also detailed more in the results and discussion how such interactions between microbes, with or without the presence of consumers can be the underlying cause of changes in soil microbial community composition, and soil aggregation.

(1) Overall so far it can be said that something happened with predators but I doubt we can really link it to the suggested feeding interactions or energy channels. Please adjust and ideally follow up with some confirmative experiments. (2) We agree that our interpretation of the results was too simplistic in some parts of the manuscript. (3) We added information about feeding preference of A. castellanii and H. nitidus, notably related to their feeding on P. fluorescens and C. globosum, respectively. For the bacterial system, we stress that effects of A. castellanii on soil aggregation were likely to be more closely related to the production of bacterial defense in response to the attack from A. castellanii than its consumption itself. For the fungal system, we stress that both changes in fungal biomass and bacterial community composition likely were important drivers of soil aggregation.

(1) Please report more information on specific PLFAs to provide an overview on other bacteria that were potentially contaminating the system. The focus is only on some aspects in the collembola treatment. (2) We agree that a more in-depth description of the microbial community was needed. (3) In the results, we added the variations of Gram+ bacteria, as well as the F : B ratio and the Gram+ / Gram– ratio. In addition, we used as well the PLFA 15:0 as general bacterial marker.

(1) According to the analyses of gram negative bacteria, it seems that these increase
with protists potentially suggesting that the inoculated Pseudomonas are not preyed upon. Please clarify and, as mentioned above, provide an experiment to show successful predation of amoeba on the exact Pseudomonas strain used here (please also report this one). (2) We agree that the lack of decrease in Gram – bacteria when the amoebae were added suggests that P. fluorescens was not or little consumed by A. castellanii. (3) The fact that P. fluorescens is a non-preferred prey for A. castellanii is thoroughly discussed in the revised version of the manuscript. We refer to the study of Jousset et al. (2009) investigating predation of A. castellanii on P. fluorescens. Jousset, A., Rochat, L., Péchy-Tarr, M., Keel, C., Scheu, S., & Bonkowski, M. (2009). Predators promote defence of rhizosphere bacterial populations by selective feeding on non-toxic cheaters. The ISME journal, 3(6), 666-674.

(1) Regarding the discussion, please adjust based on the changes in the results and comments above as I think many claims cannot be made with the current results (e.g.2nd line in discussion and the following- protists do not show a significant effect on aggregation). Similarly, a major discussion in the fungal system is based on a non significant effect. As long as we use post-hoc statistics, this is determines our findings and what we should be reported. A repetition of the experiment with more replications could be done to increase the statistical power. Overall, throughout the discussion, conclusions and abstract, many major points are not supported by results and should be adjusted. In fig.1 it is shown that there is no significant difference in the aggregate stability in bacteria alone and bacteria with protist treatments. The same holds for the fungal treatment. As such, this should not be reported as an effect in several parts of the text! (2) We agree that A. castellanii and H. nitidus induced significant changes on soil aggregate formation, but not on soil aggregate stability as indicated by direct comparison of the treatments with P. fluorescens or C. globosum with and without their associated predators. However, we observed that the significant increase in soil aggregate stability in response to the addition of P. fluorescens and C. globosum vanished when their associated predators were added. Overall, this indicates that A. castellanii and H. nitidus weakly reduced the positive effect of P. fluorescens and C. globosum

on soil aggregate stability. (3) We adjusted our description of results, as well as the discussion to underline that the reduction of soil aggregate stability is only a trend. We highlighted the stronger effect of soil aggregate formation in the conclusion as well.

(1) I wonder if the composition analysis done in Fig.3 makes sense. . . In the experiment you would expect a single PLFA marker in the fungal treatment (E.coli) and two in the bacterial treatment (E.coli and Pseudomonas- or even one as both are gram negative). (2) Bacteria do have a number of PLFAs including specific and non-specific. This is shown e.g. for Bacillus and Pseudomonas (Ruess et al. 2005, Ecology 86, 2075-2082). PLFA markers are thus not specific enough to trace specific microbial strains. Therefore, it is not possible to use PLFA markers to specifically trace E. coli and P. fluorescens. Ruess, L., Schütz, K., Haubert, D., Häggblom, M. M., Kandeler, E., & Scheu, S. (2005). Application of lipid analysis to understand trophic interactions in soil. Ecology, 86(8), 2075-2082.

(1) This figure shows that the setup seems very contaminated which make all results obtained little reliable. Also, I wonder what the relevance for these super simplistic approaches are as many things like interactions that cause biofilm production or integration of bacteria and fungi (and algae) into more stable structures might be needed to make ecological sense of aggregate formation? (2) We agree that presenting the trophic interactions as one predator (protist or collembolan) feeding on one microbial strain (P. fluorescens or C. globosum) was confusing. We did not set-up sterile microcosms, meaning that there was a residual (because of autoclaving) microbial background present in the microcosms. Although our systems are simplified, we argue that the presence of such microbial background in fact helps to link our results to more realistic and complex conditions. (3) We made clearer that the community consumed by A. castellanii and H. nitidus were not composed only of the added strains, but also included the residual microbial background present in the microcosms. We described and discussed how inoculation steps modified the microbial community, and how this in turn can be linked to soil aggregation.

(1) Please update the references and include some more recent references as only few are from 2018 and none from 2019 or 2020. A lot of work on soil biodiversity, especially on protists including interactions with bacteria, has been done in the last years- even including some papers by the authors that would bring the writing into a more novel context. (2) We agree that recent studies investigated effects of higher trophic levels on microbial communities and also on the effects of microbes on soil aggregation and that the manuscript would greatly benefit from these recent research inputs. (3) We linked our work to the recent studies linking higher trophic levels to soil microbial communities. We added most of the references suggested by reviewer 2, as well as others, such as Thakur and Geisen (2019), Lehmann et al. (2020) and Coulibaly et al. (2019). Coulibaly, S.F.M., Winck, B.R., Akpa-Vinceslas, M., Mignot, L., Legras, M., Forey, E., Chauvat, M. 2019. Functional Assemblages of Collembola Determine Soil Microbial Communities and Associated Functions. Front. Environ. Sci. 7:52. doi: 10.3389/fenvs.2019.00052 Lehmann, A., Zheng, W., Ryo, M., Soutschek, K., Roy, J., Rongstock, R., Maaß, S., Rillig, M. C. 2020. Fungal Traits Important for Soil Aggregation. Front. Microbiol. 10:2904 Thakur, M. P., Geisen, S. 2019. Trophic regulations of the soil microbiome. Trends in Microbiology, 27(9), 771-780.

(1) Minor comments L238: was higher IN these... (2) We apologize for this typo (3) The paragraph was fully rephrased
* * *
[Figure]

**Fig. 1.** Figure 1 Effect of bacterial and fungal predator-prey inoculations on microbial biomass and composition.

---

## Author Comment (AC2) · 28 Apr 2020

We thank the reviewers for the very detailed comments which helped us to improve our manuscript. The response to comments from the two referees is organized according to the Biogeosciences's guidelines as follows: (1) comments from Referees, (2) author's response, (3) author's changes in manuscript. We did not directly quote updated text nor the exact location of the changes made in the updated manuscript as co-authors are still working on it and minor changes can still occur. According to the guidelines of Biogesociences, we submit the answer to the comments from the two referees first and then the updated manuscript.

Anonymous Referee #2

(1) General: In general I find this study to be well executed and of interest. Including trophic structure in our assessments of soils is an important and understudied topic. While most of the work is well executed, the authors spend a large portion of their discussion talking about bacterial mucosal production, but this is never actually tested. If a huge portion of the work depends on understanding how trophic structure influences bacterial mucosal production, then it would be important that this is assessed. I would be hesitant to focus so intently on this interpretation, and spend more time addressing the various components you did test. Additionally, I believe that the 13C portion of this analysis to determine differences in soil and litter derived C needs to be expanded on. This could be an important conclusion, but it is unclear how this work was done, and whether or not labeled litter was added. (2) We agree that the discussion about the mucosal effect was too much developed as we did not measure mucilage it in this experiment. We agree that the use of the two C sources by the microbes can be expanded and more detailed. (3) We reduced the discussion about mucilage production in the discussion. We expanded the part about microbial C use throughout the manuscript by integrating it in the main objectives and not only as a side aspect of the article. Please see more detailed response to the 13C part in the response to the specific comment on this measurement.

(1) Abstract: No specific comments. Intro: In general, this is a well written introduction. The authors lay out multiple factors on how microbes and mesofauna influence soil aggregation. It is at times a bit repetitive though, consistently focusing on the lack of trophic structure assessment to soil aggregation. Additionally, I believe that the authors focus on soil aggregation limits the scope of this study. The authors are assessing multiple components of the soil environment, and therefore, it would be ideal if they could expand their introduction of topics beyond soil aggregation. The authors explore the influence of trophic interactions on soil microbial community formation and on the incorporation of C and on CO2 emissions. If the authors were more concise, they would have room to include additional dimensions to their work. (2) Thank you very much for this insightful comment. We agree that the scope of our study was too narrow and

restricted to soil aggregation. There is a current trend towards microbe-centered approaches linking microbial communities to higher trophic levels (Thakur and Geisen, 2019; Coulibaly et al., 2019; Lucas et al., 2020) and our work would benefit from being presented as a contribution to this effort. We apologize for the introduction being a bit repetitive sometimes. Coulibaly, S.F.M., Winck, B.R., Akpa-Vinceslas, M., Mignot, L., Legras, M., Forey, E., Chauvat, M. 2019. Functional Assemblages of Collembola Determine Soil Microbial Communities and Associated Functions. Front. Environ. Sci. 7:52. doi: 10.3389/fenvs.2019.00052 Lucas, J. M., McBride, S. G., Strickland, M. S. 2020. Trophic level mediates soil microbial community composition and function. Soil Biol. Biochem., 143, 107756. Thakur, M. P., Geisen, S. 2019. Trophic regulations of the soil microbiome. Trends in Microbiology, 27(9), 771-780. (3) We broadened the scope of our study and included effects of trophic interactions on microbial communities, C dynamics (microbial C use, SOC concentrations and $CO_2$ emissions) and soil aggregation. We re-wrote most of the introduction, results and discussion to better balance the coverage of each of these aspects.

(1) Methods: Line 101: Can you clarify this detail a bit more? I think the point is that when you add mesofauna they introduce new microbial organisms, and to account for this you also added microbes to the control treatments, but I am not entirely clear on this detail. How did you detail the Predator associated microbiota? (2) Collembolans are not sterile, they are associated with microbiota on their body surface (Anslam et al. 2016). When we added collembolans to the microcosms, we also added the microbes that were attached to their body surface. It is possible that such microbial addition modify the soil microbial community composition. To tease apart the effects of collembolans on the soil microbial community composition due to (i) the addition of their associated microbiota and to (ii) consumptive and other non-consumptive effects, we added a microbial wash of the collembolans to the fungal treatment. By doing so, the differences between the fungal and fungal + collembolan treatment provide the effect of consumptive and non-consumptive effects not related to the transport of microbes on the body of the collembolan. The effect of the transport of microbes on the body

of collembolan is tested separately in the "collembolan wash" treatment. Anslam, S., Bahram, M., Tedersoo, L. (2016) Temporal changes in fungal communities associated with guts and appendages of Collembola as based on culturing and high-throughput sequencing. Soil Biology and Biochemistry 96, 152-159. (3) We detailed the potential consumptive and non-consumptive effects of collembolans on the microbial community in the introduction. In the discussion, we now stress how the compensatory addition of the collembolan wash allowed to trace the effect of the transport of microbes on the body surface of the collembolans.

(1) I find the 13C-12C comparison protocol confusing. Could you expand your discussion of how you are able to differentiate between soil and litter sources? In particular, how are you assessing the final amount of 13C in your soils. Are you obtaining this information from GCMS work, or are you specifically measuring them using an isotopic analysis device? Additionally, I am unclear as to how you are able to ultimate differentiate whether the 13C in you sample came from litter or soil, unless you inoculated with 13C labeled litter. (2) We apologize for the lack of clarity. We used the natural difference in 13C/12C of C4 and C3 plants to trace C in microbial PLFAs originating from soil (mainly wheat origin, C3 plant), and added chopped litter (mainly maize, C4 plant). The isotopic 13C/12C ratios of the PLFAs was measured using a trace gas chromatograph (GC; Thermo Finnigan, Bremen, Germany), equipped with a DB5-DB1 column combination (30 m and 15 m, both 0.25 $\mu$m ID, Agilent), and coupled via a GP interface to a Delta Plus mass spectrometer (Thermo Finnigan, Bremen, Germany). (3) We modified the material and methods to make clear that the different 13C/12C of C3/C4 plants (wheat/maize) was used to trace C origin in microbial communities.

(1) Results: Are the control treatments truly just E. coli? I presume that because they were made from field soil, there is also a natural microbial community. This is not necessarily a problem, but if you are labeling these as E. coli only, that may be misleading. (2) We agree that labelling E. coli was confusing and that a microbial background remained in the microcosms after autoclaving. (3) We made clearer that

the community consumed by A. castellanii and H. nitidus were not composed only of the added strains, but also included remaining microorganisms in the microcosms. We described and discussed how inoculation steps modified the microbial community, and how this in turn can be linked to soil aggregation.

(1) Line 211: Awkward phrasing, maybe adjust to "Neither soil aggregate formation nor stability differed" and break this sentence up into two different sentences. (2) We apologize for the awkward phrasing. (3) We re-wrote the entire paragraph and the sentence was deleted.

(1) Lines 210-214: This paragraph starts with fungal results, but then also addresses other treatments. Maybe split this into two paragraphs, as it is difficult to follow the portion of the results in the second half of this paragraph. (2) We agree that the description of the controls should have been separated from the effect of the fungal treatment. (3) This paragraph was fully re-written.

(1) Discussion: While PLFA is an acceptable method, its ability to measure more fine scale changes in community composition is limited. It is possible that changes did occur, but they were not obvious with PLFA analysis. (2) We agree that finer changes in microbial community composition may have occurred and could have been identified using metagenomic analyses. However, while 16S rRNA metagenomics shows higher precision in depicting changes in bacterial community composition compared to PLFAs, both methods are of similar power in linking microbial community composition to soil functioning (C and N cycles, response to land-use, etc.) (Orwin et al., (2018). Orwin, K. H., Dickie, I. A., Holdaway, R., Wood, J. R. (2018). A comparison of the ability of PLFA and 16S rRNA gene metabarcoding to resolve soil community change and predict ecosystem functions. Soil Biology and Biochemistry, 117, 27-35. (3) We added a sentence in the discussion to acknowledge for possible changes in microbial community composition not detected in PLFAs.

(1) Line 223: missing a ) (2) We apologize for the typo (3) The paragraph was fully

re-written in the revised version of the manuscript

(1) Is the collembolan species used known to also feed on bacteria? If so, how would this influence the results? (2) We know that H. nitidus can feed on bacteria, notably P. fluorescens (Pollierer et al., 2019). But when fed on bacteria, H. nitidus is not able to reproduce. This indicates that bacteria are of inferior food quality than fungi for H. nitidus. This also nicely illustrates the concept of food flexibility (Briones et al., 2018) indicating that soil animals often prefer certain food resources but also feed on other resources if their preferred food source is absent. In our case, as C. globosum (which is their preferred food source, see response to reviewer 1, Pollierer et al., 2019) is provided as food source, we expected collembolans to preferentially feed on fungi. This is consistent with the decrease in fungal PLFA markers observed when collembolans were added (Figure 1 B). Of course, this does not exclude that collembolans may also have ingested some bacteria and this at least partly may explain the observed changes in bacterial community composition when H. nitidus was added. Briones, M.J.I. (2018) The Serendipitous Value of Soil Fauna in Ecosystem Functioning: The Unexplained Explained. Front. Environ. Sci. 6:149. doi: 10.3389/fenvs.2018.00149 Pollierer, M. M., Larsen, T., Potapov, A., Brückner, A., Heethoff, M., Dyckmans, J., & Scheu, S. (2019). Compound‐specific isotope analysis of amino acids as a new tool to uncover trophic chains in soil food webs. Ecological Monographs, 89(4), e01384 (3) We specified in the material and methods the feeding preference of H. nitidus for C. globosum. In the discussion, we stress that the changes in the bacterial community composition after the addition of H. nitidus may as well have been due to consumption of certain bacteria. In addition, we link changes in bacterial community composition to changes in soil aggregation in the fungal-based system.

(1) Why are the $CO_2$ respiration amounts not mentioned throughout the study? It seems like this would be of interest considering that these metrics are often used to estimate microbial biomass. (2) We initially thought of measuring $CO_2$ emission only as a control to check that living organisms were respiring during the incubation. This

is indeed a good idea to exploit these results more in our study, especially because C use, microbial biomass and soil aggregation may relate to $CO_2$ emissions. In our case, we can't use $CO_2$ emissions to estimate the microbial biomass as our systems do not have only microbes. The $CO_2$ emitted results from the respiration of microbes, but also protists and collembolans. (3) The effect of predator-prey interactions on $CO_2$ emissions was presented in figure 3, together with the effects on SOC concentrations. We re-organized the manuscript to present and discuss the effects of microbial C use, SOC concentrations and $CO_2$ emissions together as C dynamics.

(1) Additional literature to consider including: Bradford, M.A., 2016. Re-visioning soil food webs. Soil Biology and Biochemistry 102, 1–3. Bailey, V.L., Fansler, S.J., Stegen, J.C., McCue, L.A., 2013. Linking microbial community structure to -glucosidic function in soil aggregates. The ISME journal 7, 2044. Crowther, T.W., Thomas, S.M., Maynard, D.S., Baldrian, P., Covey, K., Frey, S.D., van Diepen, L.T.A., Bradford, M.A., 2015. Biotic interactions mediate soil microbial feedbacks to climate change. Proceedings of the National Academy of Sciences 112, 7033-7038. Grandy, A.S., Wieder, W.R., Wickings, K., Kyker-Snowman, E., 2016. Beyond microbes: Are fauna the next frontier in soil biogeochemical models? Soil Biology and Biochemistry 102, 40-44. Jiang, Y., Liu, M., Zhang, J., Chen, Y., Chen, X., Chen, L., Li, H., Zhang, X.-X., Sun, B., 2017. Nematode grazing promotes bacterial community dynamics in soil at the aggregate level. The ISME Journal 11, 2705-2717. Lucas, J.M., McBride, S., Strickland, M.S.S., 2020. Trophic level mediates soil microbial composition and function. Soil Biology and Biochemistry. (2) Thank you very much for these suggestions. We carefully read these articles, which helped us to better introduce our study and discuss the results. (3) We added all these references in the revised manuscript, expect the one of Bailey et al. (2013) which considered soil aggregate as a habitat for microbes, while our focus is more on the effect of soil microbes on soil aggregation. We also added several recent references to better integrate our work into the current research linking higher trophic levels to microbial communities.

[Figure]

[Figure]

**Fig. 1.** Figure 1 Effect of bacterial and fungal predator-prey inoculations on microbial biomass and composition.

[Figure]

**Fig. 2.** Figure 3 Effect of bacterial and fungal predator-prey inoculations on C dynamics.

---

## Author Response (AR2)

Dear Editor,

Thank you for the evaluation of our revised manuscript. We conducted the minor revisions requested. Please find below a point-by-point response to the comments of the two referees. Please find the updated version without apparent track-change, followed by the revised version, with track-change. Please note that the line numbers refer to the revised version without track change. All co-authors had a final read of the manuscript and agreed with this final re-submitted version.

Best wishes,

Amandine Erktan, on behalf of the co-author team.

**Report 1 - # Referee 3**

The manuscript has been greatly improved. I am satisfied with the authors' responses to the comments. I suggest that it is ready for publication.

*We are pleased that the corrections conducted satisfied Referee # 3.*

**Report 2 - # Referee 2**

After reviewing this manuscript for a second time, I believe that the authors have done a nice job of addressing my comments. The introduction is far less repetitive and also emphasizes the broader scope of their study. I congratulate them on the hard work they have put into this revision.

*We thank Referee # 2 for the positive evaluation of our revised manuscript.*

My only main concern regarding their methodology is with regard to their CUE assessment. They attempt to discern whether C came from litter or from soil by examining the difference in 13C in C4 versus C3 plants. However, their little mixture included both C4 and C3 plants, making it difficult to decipher the source of C. I request that they clarify their methodology, or include an important caveat into their methods/results about how C could have come from litter that was attributed to soil

*We agree that a bias is possible in the estimation of the two C sources, however, this bias likely is small. As noted by # Referee 2, the presence of wheat in the chopped litter (otherwise mainly composed of maize) may introduce a bias in the estimation of the relative importance of the C sources. However, such bias can occur only if microbes preferentially feed on wheat or maize litter, resulting in under- or overestimating the proportion of the chopped litter as C source. Otherwise, the reference value for chopped litter already accounts for the mixture of wheat and maize. Notably, the $\delta^{13}C$ value of the chopped litter (13.71 ‰), is close to that of maize as C4 plant (about 11 ‰) reflecting the dominance of maize in the chopped litter. Not strictly proportional use of maize and wheat litter C by microorganisms therefore are unlikely to significantly affect our calculations. Importantly, for affecting our treatment effects on the use of litter C by microorganisms it would be necessary that microorganisms in the different treatments use wheat / maize litter C in different proportions, which is highly unlikely especially considering that the litter materials were thoroughly fragmented and the microbial communities in the different bacterial as well as fungal treatments differed little. Further, especially early during decomposition differential use of litter C*

*from maize and wheat is unlikely. In the revised version of the manuscript, we added the following sentence to point to the possible bias related to the heterogeneity in $\delta^{13}C$ signal of the chopped litter (l.198-201): "Using the mean $\delta13C$ value of litter in this calculation assumes that carbon from maize and wheat litter in the litter mixture was used indiscriminately, which we assume to be justified considering the short period of time when predominantly labile litter compounds are used; further, considering that the litter mixture comprised predominantly maize litter differential incorporation of carbon from wheat is likely to affect the calculations only little." Finally, although the calculated relative importance of C sources may include some bias, we argue that the dominant use of soil C vs. litter C cannot be the result of an artefact.*

One minor concern is that there are frequent locations where the grammar needs revising. I have tried to help point these areas out, but I may have missed a few. In general the manuscript is well written, but there are some consistent subject-verb tense issues as well as a few awkward phrasing locations.

*We apologize for the errors in the grammar. We conducted the minor revision requested and double checked the English.*

However, overall I find this to be a well designed study and of importance to the field.

*We thank Referee # 2 for the positive comments on our manuscript.*

MINOR COMMENTS:

Line 43: should be "has been" not "have been"

*Corrections have been done accordingly (l.45)*

Line 49: change "Since" to "For"

*"Since a decade" was replaced by " In the last decade" (l.51)*

Line 51: change linking to link

*Corrections have been done accordingly (l.53)*

Line 63: change to "higher contents of lipids and proteins" or "higher lipid and protein content"

*We modified to " higher lipid and protein content" as suggested (l.68-69)*

Line 80: change as well to also

*"as well" was replaced by "also" as suggested (l.81)*

Line 81: change to evidence, not evidences

*"Evidences" was replaced by "evidence" (l.82).*

Line 89: change to "has also been reported"

*"as well have been reported" was replaced by " has also been reported" as suggested (l.90)*

Line 125: were added

*"was added" was replaced by "were added" as suggested (l.127).*

Line 138-139: I feel like there is a word missing or something does not connect in these lines.

*The sentence "A. castellanii is able to feed on our model strain of P. fluorescens (Jousset et al. 2009), but prefers less toxic strains" was rephrased as follow: "The amoeba species A. castellanii was shown to feed on our model strain P. fluorescens (Jousset et al. 2009), but prefers less toxic strains" (l.139-140)*

Line 143: H. nitidus shows

*The sentence was rephrased as follow: "The collembolan species H. nitidus was chosen because of its abundance in European temperate soils (Hopkin, 1997) and as it has been shown to preferentially feed on C. globosum (Pollierer et al., 2019); feeding on C. globosum it is able to survive and reproduce." (l.142-144)*

Line 144: "when fed this" drop "on"

*"The collembolan species H. nitidus was chosen because of its abundance in European temperate soils (Hopkin, 1997) and for its appetence for C. globosum (Pollierer et al., 2019). H. nitidus is showing a strong preference for C. globosum and is able to survive and reproduce when fed on this saprotrophic fungus. " was replaced by "The collembolan species H. nitidus was chosen because of its abundance in European temperate soils (Hopkin, 1997) and as it has been shown to preferentially feed on C. globosum (Pollierer et al., 2019); feeding on C. globosum it is able to survive and reproduce." (l.142-144)*

Line 144: change as well to also

*"H. nitidus is as well" was replaced by "H. nitidus is also" as suggested (l.144).*

Line 160: You mention that the litter was a mixture of maize and wheat, though more dominated by maize. Because the agricultural soil C is also mainly from wheat, how are you able to split your understanding of whether the soil C consumption was from the litter or soil organic matter?

*Indeed, the difference in the signal from maize (C4) and wheat (C3) plant was used to calculate the relative importance of the two C sources, namely the soil C and the added chopped litter. The added litter was indeed not composed by pure maize litter. We argue, however, that this little affected our calculations. We agree with referee # 2 that there is a risk of a potential bias in the calculation of the relative importance of the two C sources because of the heterogeneity of the $\delta^{13}C$ signal of the added chopped litter, but this risk remains low (see response to the 2$^{nd}$ comment of Referee # 2). We added the following sentence to point to this potential bias: "Using the mean $\delta13C$ value of litter in this calculation assumes that carbon from maize and wheat litter in the litter mixture was used indiscriminately, which we assume to be justified considering the short period of time when predominantly labile litter compounds are used; further, considering that the litter mixture comprised predominantly maize litter differential incorporation of carbon from wheat is likely to affect the calculations only little." (l.197-200)*

Line 232: "were" captured

*"was" was replaced by "were" as suggested (l.235)*

Line 243: remove the word "only"

*"only" was removed as suggested (l.246)*

Line 250: change as well to also had

*"as well" was replaced by "also" (l.254)*

Line 253: remove as well

*"as well" was removed as suggested (l.256)*

While not a essential change (and admittedly a preference of this reviewer), it would be much better to see the actual p-values instead of > or < 0.05 values.

*We agree with Referee # 2 and modified the text to show the real P-values instead of the significance levels.*

Line 271: insert the at "in THE presence of"

*"in presence of" was replaced by "in the presence of" (l.275)*

A more general opening paragraph to the discussion would be helpful. The authors jump into results again and then focus on specific results quite quickly. I would highly recommend a summarizing paragraph that emphasizes the key results and they relevance before jumping into such detail.

*We added a general summary of the main results at the beginning of the discussion (l.331 – 344): "Our results showed that simplified trophic interactions modified microbial community composition and soil aggregation, but did not or only little affect C dynamics. Overall, the effects were stronger in the fungal-based system than in the bacterial-based system. In the latter, the inoculation of P. fluorescens as dominant bacterial strain in large drove the changes in microbial community composition, whereas the addition of the amoeba predator A. castellanii did not induced further changes, presumably because P. fluorescens is a less preferred and toxic strain for A. castellanii. However, A. castellanii enhanced the formation of soil aggregates, presumably related to changes in the production of bacterial EPS in response to the attack by A. castellanii. In the fungal-based system, conform to our expectations, the inoculation of C. globosum increased fungal biomass and the addition of the grazer H. nitidus reduced it. These variations in fungal biomass were positively related to changes in soil aggregation, suggesting a detrimental effect of collembolans on soil aggregation. Surprisingly, the inoculation of C. globosum and H. nitidus resulted in significantly modified bacterial biomass and composition, and this was related to changes in soil aggregation. Finally, in the bacterial- and fungal-based systems, soil organic matter was the dominant C source and inoculation steps only weakly modified the relative importance of soil vs. added chopped litter as microbial C source. Notably, the inoculation treatments did not significantly affect SOC concentrations*

*and CO2 emissions, suggesting that despite trophic interactions significantly modify microbial communities and soil aggregation this may not change soil C dynamics."*

Additionally, the first paragraph of the discussion is currently quite long and addresses multiple ideas. Could the authors break it up into more than one paragraph?

*We split the first paragraph in two paragraphs. The first one (l.346-351) deals with the effect of the inoculation of P. fluorescens on the microbial community. The second one (l.353-373) deals with the lack of effect of A. castellanii on the microbial community.*

Line 388: awkward phrasing in "Despite of that"

*"Despite of that" was replaced by "However". Sorry for the awkward phrasing (l.408)*

Line 406: change to "also feeding on bacteria"

*"feeding as well on bacteria" was replaced by "also feeding on bacteria" as suggested (l.426).*

Line 414: as well to also

*"as well may have modified" was replaced by "also may have modified" as suggested (l.433)*

Line 414: it has "long been assumed"

*"it has been assumed for long" was replaced by "it has long been assumed" as suggested (l.434)*

Line 416: "a" microbial wash

*"addition of microbial wash" was replaced by " addition of a microbial wash" as suggested (l.436)*

Line 432: change to "were observed"

*"Effects, however, were minor as no overall changes in bacterial C source was observed between treatments and soil C remained the main source of carbon for bacterial communities." Effects, however, were minor and did not affect bacterial C source which remained to be based mainly on soil C." (l.451)*

Table 2: Again it would be preferable to have real p-values

*We replaced the significance code by the real P-value in the table 2 and in all other tables and figures (as well as in the text).*

Figure 2b: What does NRD stand for?

*Thank you for pointing this mistake. RND meant "natural reduced diversity" in a previous version of the manuscript. We forgot to change it here. In the revised version, we now modified RDN to RMB, standing for "remaining microbial background".*